# CA1 pyramidal cell diversity is rooted in the time of neurogenesis

**Davide Cavalieri, Alexandra Angelova, Anas Islah, Catherine Lopez, Marco Bocchio[†], Yannick Bollmann[‡], Agnès Baude, Rosa Cossart\***

Aix Marseille Université, Inserm, INMED, Turing Center for Living Systems, Marseille, France

**Abstract** Cellular diversity supports the computational capacity and flexibility of cortical circuits. Accordingly, principal neurons at the CA1 output node of the murine hippocampus are increasingly recognized as a heterogeneous population. Their genes, molecular content, intrinsic morpho-physiology, connectivity, and function seem to segregate along the main anatomical axes of the hippocampus. Since these axes reflect the temporal order of principal cell neurogenesis, we directly examined the relationship between birthdate and CA1 pyramidal neuron diversity, focusing on the ventral hippocampus. We used a genetic fate-mapping approach that allowed tagging three groups of age-matched principal neurons: pioneer, early-, and late-born. Using a combination of neuro-anatomy, slice physiology, connectivity tracing, and cFos staining in mice, we show that birthdate is a strong predictor of CA1 principal cell diversity. We unravel a subpopulation of pioneer neurons recruited in familiar environments with remarkable positioning, morpho-physiological features, and connectivity. Therefore, despite the expected plasticity of hippocampal circuits, given their role in learning and memory, the diversity of their main components is also partly determined at the earliest steps of development.

**\*For correspondence:**
rosa.cossart@inserm.fr

**Present address:** [†]Biosciences Institute, Newcastle University, Newcastle, United Kingdom; [‡]Developmental Neurobiology, King's College London, London, United Kingdom

**Competing interest:** The authors declare that no competing interests exist.

## Editor's evaluation

The authors use powerful fate-mapping and complementary neuroanatomical and electrophysiological approaches to examine development of hippocampal circuits. They identify a new group of pioneer CA1 pyramidal neurons, which are born early in embryonic development, contribute to CA1 neuronal diversity, and play a role in encoding familiar environments.

## Introduction

Hippocampal circuits serve multiple complex cognitive functions including navigation, learning, and episodic memory. For this purpose, they form highly associative networks integrating external inputs conveying multi-sensory, proprioceptive, contextual and emotional information onto internally generated dynamics. For many years, in contrast to their inhibitory counterparts, principal glutamatergic cells have been treated experimentally and modeled computationally as identical twins. Hence, multiple streams of information serving a variety of functions were considered to be integrated by a uniform group of neurons. This conundrum has been recently clarified by converging results indicating that the hippocampus is in reality comprised of heterogeneous principal cell populations forming at least two distinct, nonuniform parallel circuit modules that are independently controlled and involved in different behaviors. This diversity would contribute to the computational flexibility and capacity of the hippocampal circuit (*Soltesz and Losonczy, 2018*; *Valero and de la Prida, 2018*).

At population level, it is becoming increasingly evident that most features of neuronal diversity within the CA1 output node of the hippocampus distribute in a way that matches the temporal

schedules of development across transverse and radial axes. According to early autoradiographic studies, mouse CA1 principal neurons (CA1PNs) are born between E12 and birth with a peak at E14 (*Angevine, 1965*; *Bayer, 1980*). In the radial axis, successive generations of PNs occupying the principal pyramidal layer migrate past the existing earlier born neurons thus creating layers in an 'inside-out' fashion. Therefore, superficial neurons (closer to the *stratum radiatum*) are in general born later than deep neurons (closer to the *stratum oriens*).

Heterogeneity of the morpho-physiology of CA1PNs distributes in a similar pattern. Hence, adult CA1PNs located in regions associated with a presumed earlier birthdate (i.e. CA1 deep) display a smaller HCN-mediated h-current (Ih) (*Jarsky et al., 2008*; *Lee et al., 2014*; *Maroso et al., 2016*; *Masurkar et al., 2020*), a lower membrane resistance (Rm) (*Graves et al., 2012*; *Masurkar et al., 2020*) as well as a higher excitability (*Cembrowski et al., 2016*; *Mizuseki et al., 2011*) and bursting propensity (*Jarsky et al., 2008*). The same applies to the local and long-range connectivity schemes of CA1 PNs. For example, seminal early studies noted how the order of neurogenesis in the entorhinal cortex, proceeding from lateral to medial, also strictly correlated with the order of its termination on CA1 PNs along the radial axis (*Bayer, 1980*); a property recently probed at functional level (*Masurkar et al., 2017*). Notably, throughout the hippocampal trisynaptic circuit, glutamatergic cells wire according to their time of neurogenesis, with cells born at the same time more likely to be synaptically connected one to another (*Deguchi et al., 2011*). This also applies for the mesoscopic organization of adult GABAergic inhibitory circuits. Indeed, along the radial axis of development, late born PNs and subregions are more likely to drive CA1 interneurons, while early born regions and PNs receive stronger inhibitory inputs (*Donato et al., 2017*; *Lee et al., 2014*; *Oliva et al., 2016*; *Valero et al., 2015*).

Finally, functional diversity also appears to correlate with presumed birthdate. When combining the information of many recent reports, a picture emerges by which in the dorsal hippocampus putatively early born PNs are comprised of a higher fraction of place-modulated neurons (*Danielson et al., 2016*; *Mizuseki et al., 2011*) with a poorer spatial coding specificity (*Danielson et al., 2016*; *Geiller et al., 2017*; *Hartzell et al., 2013*; *Henriksen et al., 2010*; *Oliva et al., 2016*). More particularly, in CA1, presumably older PNs are better tuned to receive external sensory inputs as their firing is more anchored to environmental landmarks. In contrast, presumably later born PNs, may preferentially convey self-referenced information, participate in SWRs and display slower if any remapping as well as more stable place maps (*Danielson et al., 2016*; *Fattahi et al., 2018*; *Mizuseki et al., 2011*; *Sharif et al., 2020*; *Valero et al., 2015*). Similarly, function segregates along the radial axis in the ventral hippocampus where deep CA1PNs specifically projecting to the nucleus accumbens shell (NAcc) or to the Lateral hypothalamus (LHA) were reported to contribute to social and anxiety-related behaviors, respectively (*Jimenez et al., 2018*; *Okuyama et al., 2016*).

Altogether, mostly based on the tight correlation between their soma position and their morpho-functional attributes, these recent results indirectly support the intriguing possibility that CA1PNs diversity may be partly determined at their time of neurogenesis. Alternatively, diversity may simply reflect final soma position and the influence of local circuits rather than an early predetermination. In order to test directly these two non-mutually exclusive possibilities, we have fate-mapped three groups of age-matched CA1PNs: pioneer (i.e. born at embryonic day 12: E12.5), early (E14.5)-, and late (E16.5)-born as described previously (*Marissal et al., 2012*; *Save et al., 2019*). We analyzed their morphophysiological properties, connectivity, and activation during the free exploration of differently familiar environments. We focused on the ventral hippocampus as this region displays the wider diversity of CA1PNs in terms of projection patterns (*Cembrowski et al., 2016*; *Jimenez et al., 2018*; *Jin and Maren, 2015*; *Kim and Cho, 2017*; *Lee et al., 2014*; *Okuyama et al., 2016*; *Parfitt et al., 2017*; *Xu et al., 2016*).

Whereas the radial position of CA1PNs correlates with some features of synaptic connectivity like the frequency of excitatory synaptic currents they receive, other morphophysiological properties including apical dendritic length in the *stratum radiatum*, parvalbumin somatic coverage, long-range projection, spiking in response to current injections, sag current or input resistance distinguish between cells with different birthdates. In particular, pioneer E12-CA1PNs stand out as a singular subset of cells regarding many parameters, broadly distributed along the radial axis and preferentially activated when mice explore a familiar environment. Therefore, the present study reveals how the heterogeneity of CA1PN diversity extends beyond the mere subdivision into two sequentially

generated sublayers along the radial axis. It likely encompasses many, at least three, intermingled subtypes specified at progenitor stage by their temporal origin.

## Results

### Pioneer CA1 pyramidal cells broadly integrate the pyramidal layer and display remarkable features

CA1 pyramidal neurons (CA1PNs) were fate-mapped using the inducible transgenic driver line *Neurog-2$^{CreER}$*, expressing CreER under the control of the Neurogenin 2 (*Neurog2*) gene promoter. We crossed *Neurog2$^{CreER}$* mice with the Ai14 reporter line (*Madisen et al., 2010*), including a CreER-dependent TdTomato (Tdt) allele (see Materials and method). Like in our previous studies (*Marissal et al., 2012*; *Save et al., 2019*), three different groups of pyramidal neurons were labeled by TdTomato expression via tamoxifen administration at separate embryonic time points: embryonic day 12.5 (E12.5), E14.5, and E16.5. We focused on the main features classically expected to segregate into two CA1 sublayers, the *deep* and *superficial* one. Note that, unless specified otherwise (*Figures 1 and 2*), all experiments were performed in the intermediate-to-ventral portion of CA1.

We first examined the somatic location and abundance of CA1PNs in the *stratum pyramidale* according to their fate-mapped date of birth *Figure 1A*. The location was calculated as the distance between the center of the soma and a line representing the *stratum radiatum/pyramidale* border. Consistent with an inside first-outside last patterning (*Angevine, 1965*; *Bayer, 1980*), we found that E16.5 CA1PNs were mostly located in the superficial part of the pyramidal layer, whereas the deep sublayer was enriched in E14.5 CA1PNs. In both dorsal and ventral CA1, E12.5 cells were also positioned in the deeper portion, although significantly shifted towards the *stratum oriens*. Noteworthy, not only did the location in respect to the *radiatum/pyramidale* border change, but also the spatial dispersion of cells in the pyramidal layer varied with the birthdate. More specifically, E12.5 were the most spread out (E12.5 interquartile range or $IQR_{E12.5}$ = 63.14 μm in dorsal and 104.44 μm in ventral CA1) and later born cells showed drastically reduced values of dispersion ($IQR_{E14.5}$ = 22 μm and 30.71 μm, $IQR_{E16.5}$ = 22 μm and 12.95 μm dorsally and ventrally, respectively). The broad distribution of E12.5 CA1PNs and its shift toward the *stratum oriens* was particularly obvious in cumulative distribution plots obtained from dorsal and ventral CA1 where a consistent fraction of the labeled E12 CA1PNs were located outside from the *stratum pyramidale*, in the *stratum oriens* and, fewer, in the *stratum radiatum Figure 1B*. In addition, they show how the deep CA1 sublayer is comprised of a mixed population of E12.5 and E14.5 CA1PNs while cells located in the *stratum oriens* are predominantly of E12.5 origin. In addition, we decided to assess the relative cell amount of the three birthdate groups of CA1 pyramidal neurons across the entire long axis. Using a combination of clarified brain samples and light-sheet microscopy (*Figure 1C*), we obtained a quantitative cartography of fate-mapped PNs according to the Allen brain atlas registration. We calculated the numbers of E12.5 (n = 4 brains), E14.5 (n = 3), and E16.5 (n = 3) PNs in the whole CA1 area and found that the number E14.5 CA1PNs per brain (6810 neurons ± 3454; mean ± SD) was more than seven times higher than E12.5 CA1PNs (917 neurons ± 851) and almost two times larger than E16.5 CA1PNs (3585 ± 401). These results are globally in line with previous work (*Angevine, 1965*; *Bayer, 1980*). However, they indicate how not only the location, but also the abundance and soma dispersion in the pyramidal layer depend on the birthdate.

We next investigated the diversity among CA1PNs using two classical immunohistochemical markers used in several previous studies to distinguish the deep and superficial sublayers. Indeed, parvalbumin (PV) was demonstrated to preferentially decorate the somata of deep CA1PNs, indicating a stronger innervation by putative PV-basket cells (*Lee et al., 2014*; *Soltesz and Losonczy, 2018*; *Valero et al., 2015*), while the calcium-buffer calbindin (CB) is mainly expressed by superficial CA1PNs (*Valero et al., 2015*; *Li et al., 2017*; *Baimbridge et al., 1991*). We expected to find preferential PV innervation and low CB expression in E12.5 and E14.5 PNs, both located in the deep sublayer. Surprisingly, these markers were better segregating according to birthdate than position. Indeed, E12.5 CA1PNs displayed significantly less putative PV contacts (see *Figure 2A*) and higher CB expression (see *Figure 2B*) than E14.5 CA1PNs, even though their somata were mainly located in the deep CA1. These first observations indicate that, at single-cell level, birthdate may better correlate with the

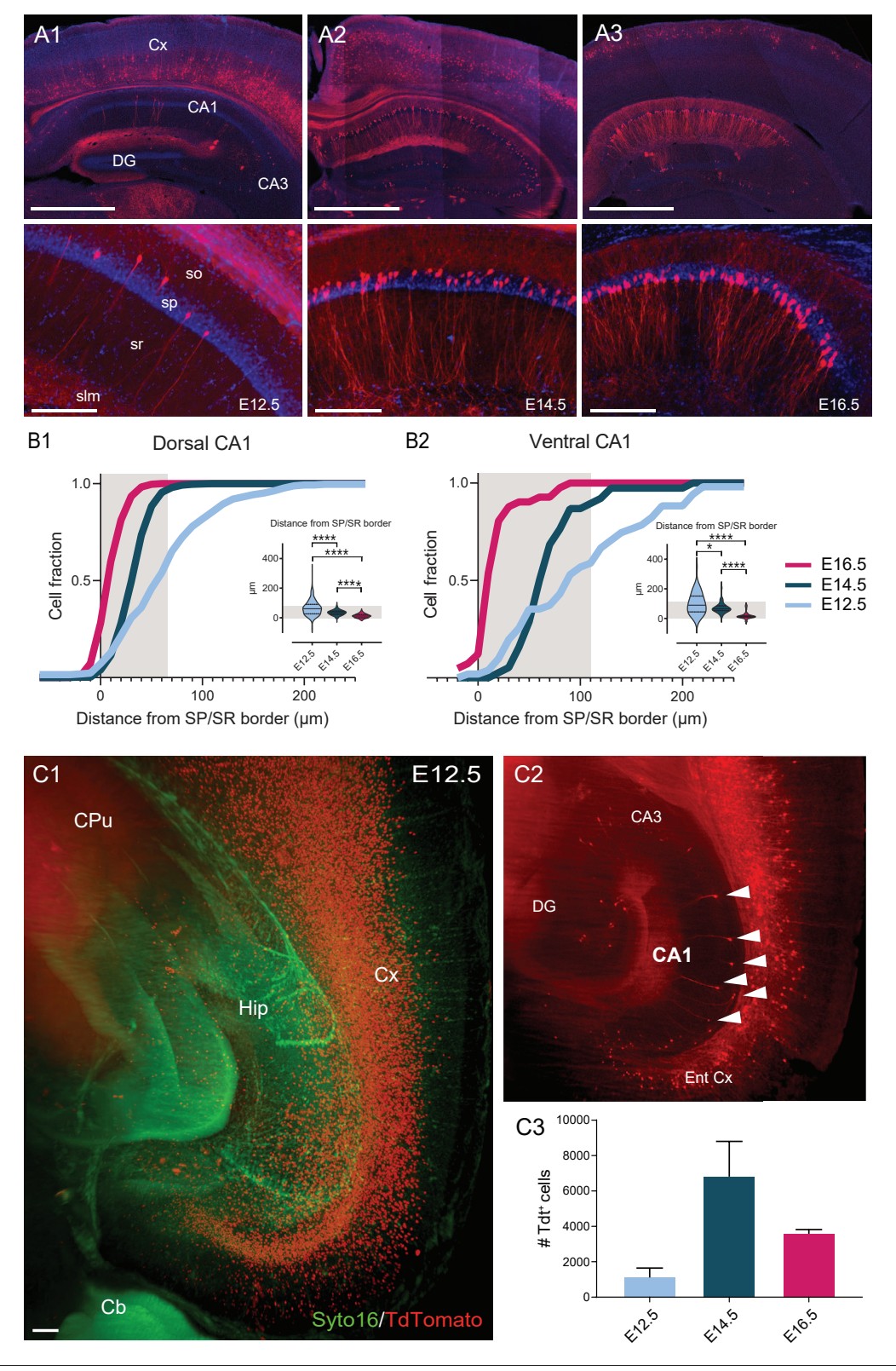

**Figure 1.** Soma location distribution of CA1PNs changes according to birthdate. (**A1-3**) Top, representative sections of the dorsal hippocampus and cortex, illustrating Tdtomato (Tdt) labeling in CA1 pyramidal neurons (PNs) in *Neurog2^{CreER}*-Tdt mice after tamoxifen induction at embryonic day 12.5 (E12.5), E14.5, and E16.5, from left to right. Bottom, higher magnification on CA1. E12.5 PNs are rare and dispersed, while E14.5 and E16.5 are

*Figure 1 continued on next page*

*Figure 1 continued*

predominantly found in the deep (upper) and superficial (lower) portion of the pyramidal layer, respectively. Note that in the above cortical areas, E12.5 PNs are restricted to the deepest layers, E14.5 PNs the middle ones (IV-III) and E16.5 PNs are most superficial (layers I-II). Scalebars: top, 1000 µm; bottom, 200 µm. Cx: Cortex; DG: dentate gyrus; so: stratum oriens; sp: stratum pyramidale; sr: stratum radiatum; slm: stratum lacunosum-moleculare. (**B1-2**) Quantification of the soma location distribution of fate-mapped CA1 PNs. Gray shaded areas represent the thickness of the stratum pyramidale. (**B1**) Cumulative fraction of E12.5, E14.5 and E16.5 PNs in dorsal CA1, calculated as distance in µm from the superficial (lower) border of the stratum pyramidale. Inset, E12.5 PNs are located further away from the border than E14.5PNs (p<0.0001, CI$_{95\%}$ [18.18; 33]) and E16.5 PNs (p<0.0001, CI$_{95\%}$ [41; 55.73]). In turn, E14.5PNs also occupy deeper positions than E16.5 PNs (p<0.0001, CI$_{95\%}$ [21; 25]). (**B2**) Cumulative fraction of E12.5, E14.5, and E16.5 PNs in ventral CA1. Inset, similarly to dorsal CA1, in ventral CA1 E12.5 PNs are located deeper than E14.5 PNs (p: 0.0175, CI$_{95\%}$ [3.13; 57.72]) and E16.5 PNs (p<0.0001, CI$_{95\%}$ [53.93; 109.57]), and E14.5 PNs than E16.5 PNs (p<0.0001, CI$_{95\%}$ [44.47; 59.32]). (**C1-2**) Quantification of cell abundance of fate-mapped CA1PNs. (**C1**) Two-dimensional view of the 3D reconstruction of a clarified hemisphere obtained by light-sheet microscopy from a *Neurog2^CreER*-Tdt mouse induced at E12.5. In red, fate-mapped glutamatergic neurons within hippocampus (Hip) and deep layers of cortex (Cx). Green fluorescent protein Syto16 is used to visualize brain structures. Cb, cerebellum; CPu, caudate putamen; Hip, hippocampus; Cx, cortex; scale bar: 100 µm. (**C2**) Optical slice extracted from the same brain as in C1, with focus of hippocampal CA1 area. TdTomato-expressing PNs in CA1 are indicated by white arrows. Ent Cx, entorhinal cortex; DG, dentate gyrus. C3 Histogram representing the abundance of fate-mapped glutamatergic pyramidal neurons in the whole CA1 area (mean ± SD).

The online version of this article includes the following figure supplement(s) for figure 1:

**Source data 1.** Source data for cell distribution of fate-mapped PNs in dorsal and ventral CA1.

**Source data 2.** Source data for cell abundance of fate-mapped CA1 PNs from whole CA1 in clarified brains.

properties of CA1PNs than position along the radial axis. They also point-out at the pioneer subset of CA1PNs as potential outliers in the relationship between cell diversity and birth order.

## Synaptic input drive onto CA1PNs reflect both radial positioning and birthdate

To further analyze the relationship between birthdate, cell position and single-cell morpho-physiological features, we next performed a series of ex vivo whole cell patch clamp recordings in acute brain slices from adult *Neurog2^CreER*-Tdt mice (E12.5: n = 12 CA1PNs from 6 mice, E14.5: 15 CA1PNs from 7 mice, E16.5: 13 CA1PNs from 7 mice), sampling from intermediate to ventral CA1.

Previous studies suggest that cells located in the deep or superficial portion of the CA1 pyramidal layer, differ in their synaptic inputs (*Kohara et al., 2014*; *Lee et al., 2014*; *Masurkar et al., 2017*; *Valero et al., 2015*). In order to investigate whether this pattern would also depend on birthdate we have voltage-clamped the fate-mapped CA1PNs at the reversal potential for GABAergic and glutamatergic currents to record spontaneous excitatory and inhibitory postsynaptic currents (s-EPSCs and s-IPSCs, *Figure 3*). The location of the neurobiotin-filled somata of the recorded cells was measured in respect to the depth of the pyramidal layer and expressed as a ratio between 0 and 1, representing the border with the *stratum radiatum* and the *stratum oriens,* respectively. Although the amplitude and frequency of neither s-EPSCs (*Figure 3C*) nor s-IPSCs (*Figure 3D*) differed, the excitatory/inhibitory balance (computed as the E/I amplitude ratio, *Figure 3E*) was lower for E14.5 than both E12.5 (p: 0.020) and E16.5 CA1PNs (p: 0.018). This indicates that neurons born at E14.5 are subject to a synaptic drive leaning more toward inhibition that the other two groups, in agreement with their receiving more putative perisomatic PV contacts (see above). In contrast, the frequency of s-EPSCs (*Figure 3C*) showed a linear correlation with the cell body location (r = 0.49, p: 0.002). This was also reflected in the E/I frequency ratio (r = 0.43, p: 0.0047), indicating a higher excitatory synaptic drive in deep than in superficial neurons, regardless of their birthdate (*Figure 3E*).

To acquire further understanding of the different connectivity profiles, we applied brief electric stimulations in the *stratum radiatum*, which contains CA3 Schaffer collaterals innervating CA1, and recorded evoked PSCs in voltage-clamp mode (*Figure 4A*). In contrast to *Valero et al., 2015*, we could not observe any correlation between the properties of evoked synaptic excitation and the soma location or the birthdate (*Figure 4B&C*). This contrasted with evoked inhibitory responses (*Figure 4B&D*). Indeed, evoked IPSC amplitude increased linearly toward the *stratum oriens* (r = 0.34, p: 0.0128), while IPSC kinetics (half-width and time constant) inversely correlated with the location (respectively,

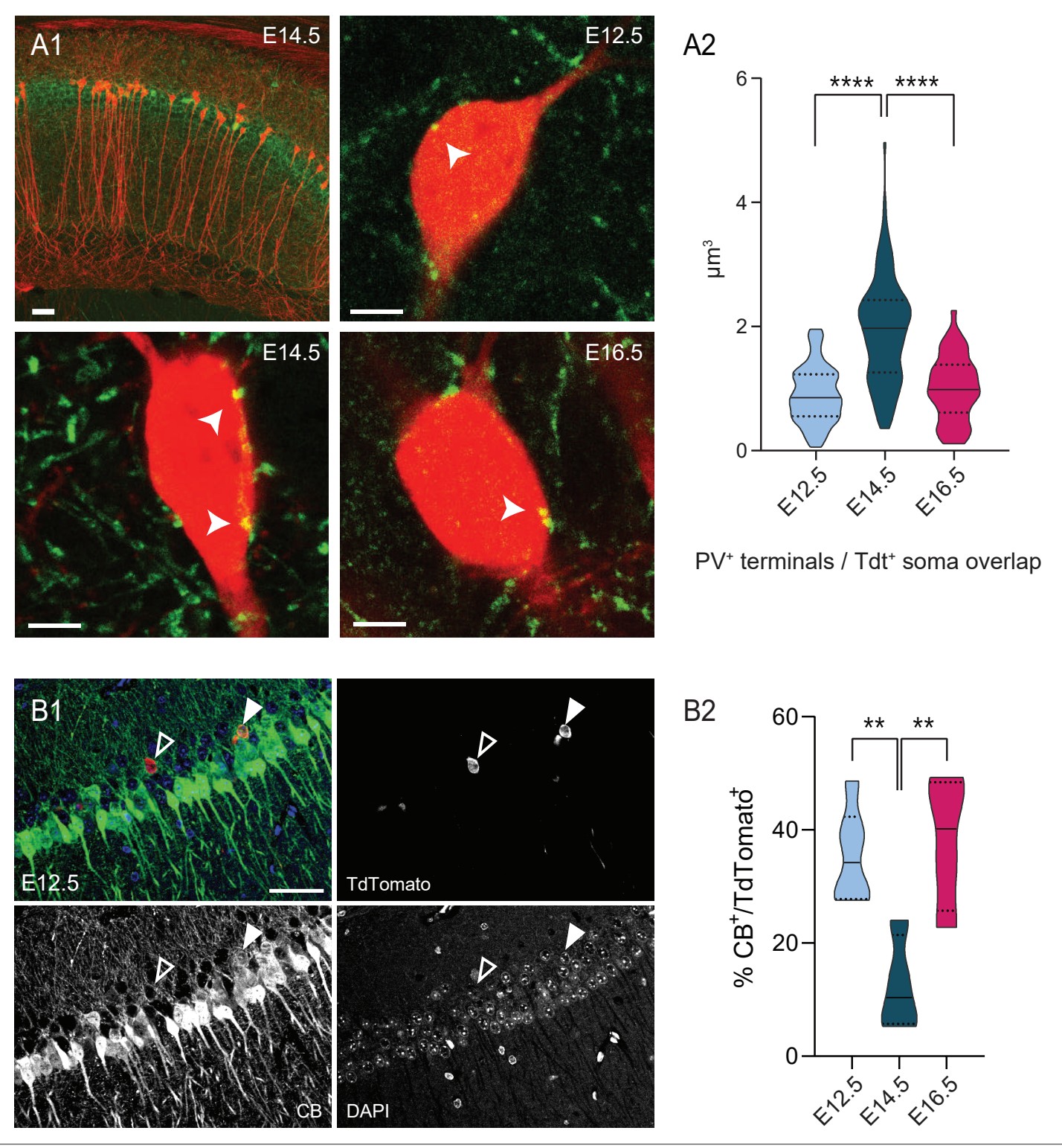

**Figure 2.** PV innervation and CB expression are biased by birthdate for pioneer cells. (**A1**) Top left, representative view of the dorsal CA1 area in a *Neurog2^CreER*-Tdt mouse tamoxifen-induced at E14.5, with Tdt+ cells (red) and PV cells (green). Top right, close-up on the soma of E12.5 PNs (red) displaying putative somatic PV boutons (indicated by a white arrow). Bottom left, close-up on E14.5 PNs. Bottom right, close-up on E16.5 PNs. (**A2**) Quantification of the volume overlap (μm³) of putative PV terminals with Tdt+ soma depending on the birthdate in dorsal CA1. E14.5 PNs present more putative boutons than E12.5 ($p < 0.0001$, $CI_{95\%}$ [–1.264; –0.81]) and E16.5 ($p < 0.0001$, $CI_{95\%}$ [0.66; 1.22]), while the latter two are not significantly different ($p$: 0.7038, $CI_{95\%}$ [–0.389; 0.19]). Scalebars: top left: 50 μm; top right, bottom left and bottom right: 5 μm. (**B1**) Representative view of a dorsal CA1

*Figure 2 continued on next page*

*Figure 2 continued*

section, after immunohistochemical staining for CB in a *Neurog2^CreER*-Tdt mouse tamoxifen-induced at E12.5 (merge, top left), with Tdt$^+$ cells (red, top right) and CB (green, bottom left) and DAPI (blue, bottom right). The black and white arrow indicate a CB-negative and a CB-positive E12.5 CA1PNs, respectively. Scalebar: 50 μm. (B2) Percentage of CB-expressing Tdt$^+$-PNs depending on the birthdate in CA1. Both E12.5PNs (p:0.0084, CI$_{95\%}$ [6.39; 39.5]) and E16.5 PNs (p: 0.0075, CI$_{95\%}$ [–43.7; –7.4]) have a higher CB expression rate than E14.5 PNs, while no difference is detected between E12.5 and E16.5 PNs (p:0.90). One-way ANOVA with Tukey's post hoc test, F (2, 11) = 9.2091; p:0.0045. Violin plots present medians (center), interquartile ranges (bounds), minima and maxima. Color-code: E12.5: light blue, E14.5: dark blue, E16.5: magenta. **p < 0.01, ****p < 0.0001.

The online version of this article includes the following source data and figure supplement(s) for figure 2:

**Source data 1.** Source data for putative PV bouton volume on fate-mapped PNs in dorsal CA1.

**Source data 2.** Source data for CB expression in fate-mapped CA1 PNs in whole CA1.

*r* = −0.36, p: 0.008 and *r* = −0.36, p: 0.0051, *Figure 4D*). This suggests that cells in the deep sublayer receive stronger and faster presumably CA2/CA3-driven GABAergic input (*Lee et al., 2014*). Finally, we tested paired-pulse (PP) responses as a proxy of the input release probability and found that inhibitory PP ratio was consistently higher in E12.5 CA1PNs as compared to E14.5 (p: 0.0326) and E16.5 (p: 0.0177) CA1PNs (*Figure 4D*). It should be noted that the presence of two ectopic cells located in the *stratum radiatum* within our E12.5 CA1PN sample does not bias our findings. Indeed, their exclusion from the analysis did not alter any linear relationship between synaptic signaling properties and cell location shown in *Figures 3 and 4* (see *Figure 4—source data 2*).

Altogether these results indicate that input connectivity motifs might be expressed as a gradient through the depth of the pyramidal cell layer. However, they also show that some functional features, such as the E/I balance and the SC-associated GABAergic release probability, may rather be established through developmental programs that are revealed uniquely by the temporal origin.

## Relationship between birthdate, soma position, and synaptic output

We have seen above that the radial position of CA1PNs is partly reflected in their synaptic input drive. Given that these cells were shown to be diverse regarding their projection area, we decided to test whether birthdate could segregate CA1PNs with different projection patterns. The ventral CA1 is known to display multiple projections targeting the EC, the amygdala (Amy), nucleus accumbens (NAcc), the medial prefrontal cortex (mPFC), the lateral septum (LS), and lateral hypothalamus (LHA) (*Arszovszki et al., 2014*; *Cenquizca and Swanson, 2007*; *Ciocchi et al., 2015*; *Kim and Cho, 2017*; *van Groen and Wyss, 1990*). Interestingly, an anatomical segregation in a laminar fashion was often reported when comparing cells with different projections.

To this aim, we performed injections (total n = 52 *Neurog2^CreER*-Tdt mice) of the retrograde tracer cholera toxin subunit B (Ctb) Ctb-647 into the Amy (E12.5: n = 3 mice, E14.5: n = 4, E16.5: n = 4), NAcc (E12.5: n = 2, E14.5: n = 4, E16.5: n = 4), mPFC (E12.5: n = 2, E14.5: n = 5, E16.5: n = 4), LS (E12.5: n = 3, nE14.5: n = 6, E16.5: n = 4) and LHA (E12.5: n = 2, E14.5: n = 3, E16.5: n = 2) and counted Ctb$^+$-Tdt$^+$ co-labeled cells in the ventral CA1 (*Figure 5*, and *Figure 5—figure supplements 1 and 2*). First, we compared the mean fraction of Ctb$^+$-Tdt$^+$ per animal among the three birthdate groups (E12.5: n = 9 mice, E14.5: n = 18, E16.5: n = 14). To focus on the overall rate of projection, we first excluded from each birthdate group the region where they projected the most densely (LS for E12.5 and E16.5 CA1PNs, NAcc for E14.5 CA1PNs). We found that co-labeled cells were significantly more abundant in E12.5 than E16.5 (p < 0.05) while no significant difference was observed between E12.5 and E14.5 (p = 0.070, *Figure 5B*). Next, co-labeling data from a given animal were translated into a binary vector of length N equal to the total amount of Tdt$^+$-birthdated cells, with '1' entries reporting the number of those that were also positive for Ctb. Vectors representing data from animals labeling CA1PNs with the same birthdate and injected with Ctb in the same target were pooled together in a 'group vector'. Different group vectors were next compared in a pairwise-fashion using a resampling approach (see Materials and methods). Note that for simplicity, in *Figure 5C and D* each group vector is represented by datapoints indicating the fraction of Ctb$^+$-Tdt$^+$ cells per animal within the group. Using this approach, we discovered that E12.5 neurons projected homogeneously to all structures analyzed, while marked differences in output innervation were found within E14.5 and E16.5 neurons (*Figure 5C*). In addition, E14.5 CA1PNs showed a clear bias toward the NAcc, followed by LS and mPFC. Similarly, and despite some variability, LS and NAcc were also preferred outputs of E16.5 CA1PNs, while the remaining three structures (Amy, mPFC, LHA) presented little to no co-labeling.

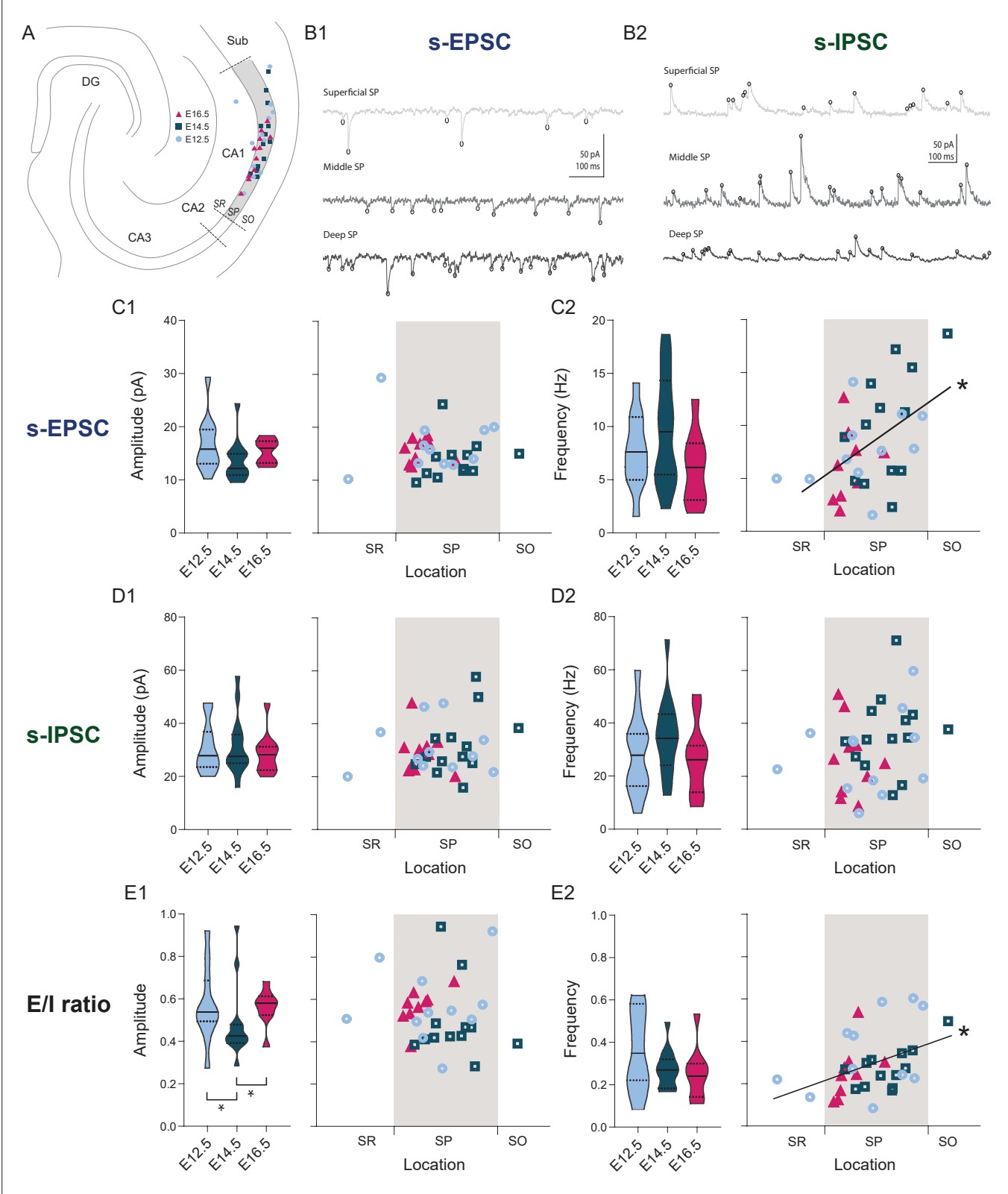

**Figure 3.** Overall synaptic drive is defined by the time of neurogenesis. (**A**) Anatomical location of neurobiotin-filled CA1PNs recorded from acute horizontal slices in voltage-clamp experiments. DG: dentate gyrus, Sub: subiculum, SR: stratum radiatum, SP: stratum pyramidale, SO: stratum oriens. (**B1–B2**) Representative traces of spontaneous excitatory synaptic currents (s-EPSCs) and inhibitory synaptic currents (s-IPSCs) from CA1PNs located respectively in the deep, middle, and superficial portion of the pyramidal layer (SP). Black circles represent detected synaptic events. Note that the

*Figure 3 continued on next page*

*Figure 3 continued*

occurrence of s-EPSCs increases from top (superficial) to bottom (deep), whereas no apparent change in s-IPSC frequency can be identified among different locations. (**C1**) s-EPSC amplitude recorded in E12.5, E14.5, and E16.5 CA1PNs. On the left panel, violin plot of E12.5, E14.5, and E16.5 PNs s-EPSC amplitude (E12.5 vs E14.5, $P_{adj}$: 0.3228; E12.5 vs E16.5, $P_{adj}$:0.4012; E14.5 vs E16.5, $P_{adj}$:0.3228). On the right panel, color-coded scatterplot of the same data plotted against the radial position of each cell. No linear correlation between s-EPSC amplitude and location (p: 0.4495,$CI_{95\%}$ [–0.45; –0.44]) was found. (**C2**) s-EPSC frequency recorded in E12.5, E14.5, and E16.5 CA1PNs. On the left panel, violin plot of the three birthdate groups (E12.5 vs E14.5, $P_{adj}$:0.3386, E12.5 vs E16.5, $P_{adj}$:0.4845, E14.5 vs E16.5, $P_{adj}$:0.4845). On the right panel, scatterplot of the same data against the radial position. S-EPSC frequency increases linearly with the cell location, the two being positively correlated (p: 0.017; $CI_{95\%}$ [0.186; 0.710]). The deeper, the more frequent spontaneous excitatory events a cell receives. (**D1**) s-IPSC amplitude recorded in E12.5, E14.5 and E16.5 CA1PNs. On the left panel, violin plot of the three birthdate groups (E12.5 vs E14.5, $P_{adj}$:0.8238; E12.5 vs E16.5, $P_{adj}$>0.999; E14.5 vs E16.5, $P_{adj}$>0.999). On the right panel, scatterplot of the same data against the radial position. No linear correlation between s-IPSC amplitude and location was found (p: 0.12, $CI_{95\%}$ [–0.177; –0.492]). (**D2**) s-IPSC frequency recorded in E12.5, E14.5, and E16.5 CA1PNs. On the left panel, violin plot of the three birthdate groups (E12.5 vs E14.5, $P_{adj}$:0.282; E12.5 vs E16.5, $P_{adj}$:0.4526; E14.5 vs E16.5, $P_{adj}$:0.1575). On the right panel, scatterplot of the same data against the radial position. No linear correlation between s-IPSC frequency and location was found (p: 0.067, $CI_{95\%}$ [–0.019; –0.485]). (**E1**) Ratio between EPSC and IPSC (E/I) amplitude recorded in E12.5, E14.5, and E16.5 CA1PNs. On the left panel, violin plot of the three birthdate groups. E14.5 PNs display a lower ratio than E12.5 PNs ($P_{adj}$: 0.010; $CI_{95\%}$ [0.031; 0.210]) and E16.5 PNs ($P_{adj}$: 0.006; $CI_{95\%}$ [–0.193; –0.057]), suggesting that their overall synaptic drive leans more toward inhibition than the other two groups (E12.5 vs E16.5, $P_{adj}$:0.2061). On the right panel, scatterplot of the same data against the radial position. No linear correlation between E/I amplitude ratio and location was found (p:0.4615, $CI_{95\%}$ [–0.38; 0.33]). (**E2**) E/I frequency ratio recorded in E12.5, E14.5, and E16.5 CA1PNs. On the left panel, violin plot of the three birthdate groups (E12.5 vs E14.5, $P_{adj}$: 0.61, E12.5 vs E16.5; $P_{adj}$: 0.61; E14.5 vs E16.5, $P_{adj}$: 0.59). On the right panel, scatterplot of the same data against the radial position. E/I ratio linearly correlates with the cell location, similarly to s-EPSC frequency (p: 0.0043; $CI_{95\%}$ [0.126; 0.654]). Violin plots present medians (center), interquartile ranges (bounds), minima and maxima. Color-code: E12.5: light blue, E14.5: dark blue, E16.5: magenta. The gray shaded area in scatterplots represents the thickness of the stratum pyramidale. *p < 0.05.

The online version of this article includes the following figure supplement(s) for figure 3:

**Source data 1.** Source data for spontaneous synaptic currents of fate-mapped CA1 PNs.

The overall higher Ctb$^+$-Tdt$^+$ fraction in E12.5 PNs (*Figure 5B*) was found as well when looking within given target structures (*Figure 5D*). Indeed, E12.5 CA1PNs projected proportionally more than E16.5 CA1PNs to Amy, and more than both E14.5 and E16.5 CA1PNs to mPFC. In contrast, within NAcc-projecting neurons, Ctb$^+$-Tdt$^+$ proportion was the greatest among E14.5 neurons, as compared to E12.5 and E16.5. In addition, E14.5 cells projected more to Amy than their later-generated counterparts (*Figure 5D*). These results are in line with previous findings that NAcc, Amy and mPFC are preferentially targeted by earlier-generated neurons located in the deep sublayer (*Jimenez et al., 2018*; *Lee et al., 2014*; *Okuyama et al., 2016*). Furthermore, they point at a specific contribution of the embryonic origin in defining output connectivity motifs (e.g. NAcc-projecting neurons mainly identified among E14.5 CA1PNs). In addition, the tendency of E12.5 CA1PNs to contact at consistent rates all retro-traced regions is reminiscent of multiple projecting CA1 cells (*Ciocchi et al., 2015*; *Gergues et al., 2020*).

## Intrinsic electrophysiological properties of CA1PNs correlate with birthdate

Together with synaptic input connectivity, intrinsic electrophysiological properties are known to contribute to the specific activation of CA1PNs, for example in the formation of place fields (*Bittner et al., 2017*; *Epsztein et al., 2011*). They have been shown to vary across cells (*Dougherty et al., 2012*; *Graves et al., 2012*; *Maroso et al., 2016*; *Masurkar et al., 2020*; *Mizuseki et al., 2011*). We have therefore examined their relationship with birthdate by performing current-clamp experiments in adult slices where CA1PNs from the three age groups could be identified (*Figure 6*; E12.5: n = 10 CA1PNs from 6 *Neurog2$^{CreER}$*-Tdt mice, E14.5: n = 12 CA1PNs from 6 mice, E16.5: n = 10 CA1PNs from 3 mice).

We first probed the intrinsic membrane excitability via a series of hyperpolarizing current steps (*Figure 6C*) and found that pioneer E12.5 CA1PNs exhibited a reduced repolarizing sag response (E12.5 vs E14.5, p: 0.0078), and rebound potential (E12.5 vs E14.5, p:0.0063). On the contrary, pyramidal neurons born at E14.5 had a higher input membrane resistance (Rm, *Figure 6B*) and, upon depolarizing current injections of increasing amplitude, were globally able to trigger more action potentials (AP) than the earlier- and later-born counterparts (*Figure 6D*, main effect of birthdate: p < 0.0001, E12.5 vs E14.5: p < 0.0001, E14.5 vs E16.5: p < 0.0001, E12.5 vs E16.5: p = 0.645, two-way ANOVA). We did not find any difference in AP threshold, half-width, after hyperpolarization (AHP) or

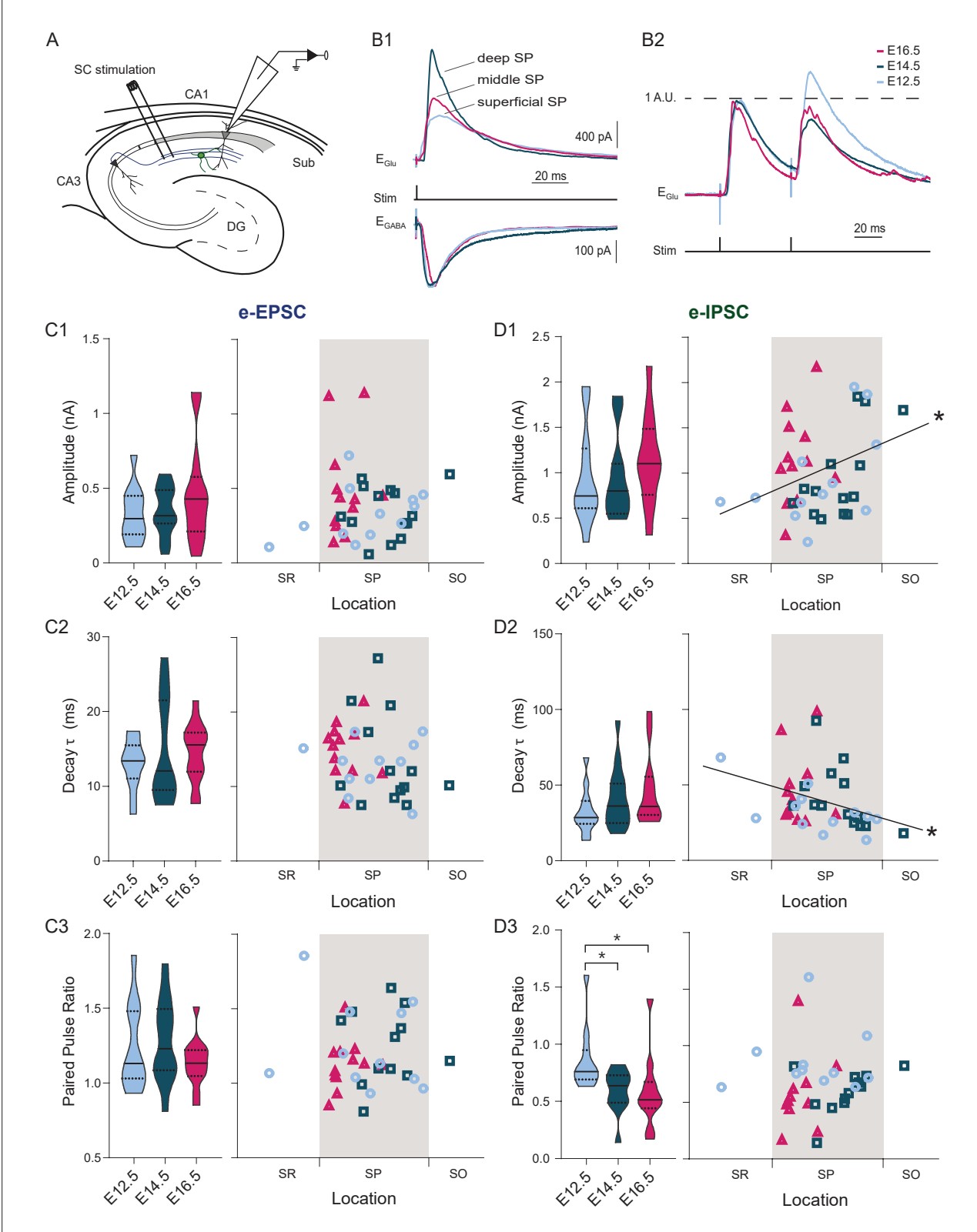

**Figure 4.** Both developmental and positional factors determine Schaffer collateral-evoked inhibition. (**A**) Experimental paradigm in which synaptic responses to Schaffer collaterals (SC) stimulation are recorded from CA1PNs. DG: dentate gyrus, Sub: subiculum. (**B1**) Representative mean traces of evoked inhibitory (top) and excitatory (bottom) synaptic currents (e-IPSCs and e-EPSCs) recorded upon electric stimulation in the stratum radiatum from CA1PNs, located respectively in the deep, middle, and superficial portion of the pyramidal layer (SP). Note that the amplitude and kinetics of e-IPSCs

*Figure 4 continued on next page*

*Figure 4 continued*

vary from deep to superficial. (**B2**) Representative mean traces of paired pulse response recorded at glutamatergic receptor reversal potential ($E_{Glu}$) and normalized on the first pulse amplitude. The current response to the second pulse in the E12.5PNS shown is proportionally larger than in E14.5 and E16.5 CA1PNs. (**C1-3**) e-EPSC amplitude (**C1**, E12.5 vs E14.5, $P_{adj}$:0.617; E12.5 vs E16.5, $P_{adj}$:0.5985; E14.5 vs E16.5, $P_{adj}$:0.617), decay $\tau$ (**C2**, E12.5 vs E14.5, $P_{adj}$:0.6165; E12.5 vs E16.5, $P_{adj}$:0.4648; E14.5 vs E16.5, $P_{adj}$:0.6165), and paired pulse ratio (**C3**, PPR; E12.5 vs E14.5, $P_{adj}$:0.916; E12.5 vs E16.5, $P_{adj}$:0.718; E14.5 vs E16.5, $P_{adj}$:0.9165) recorded in E12.5, E14.5 and E16.5 CA1PNs. Left, violin plot of the three birthdate groups. Right, scatterplot of the same data against the radial position. No linear correlation with soma location was found in any of the three measures (e-EPSC amplitude, p:0.126; decay, 0.1315; PPR, 0.4412). (**D1–D3**) e-IPSC amplitude (**D1**, E12.5 vs E14.5, $P_{adj}$:0.418; E12.5 vs E16.5, $P_{adj}$:0.222; E14.5 vs E16.5, $P_{adj}$:0.222), decay $\tau$ (**D2**, E12.5 vs E14.5, $P_{adj}$:0.4018; E12.5 vs E16.5, $P_{adj}$:0.375; E14.5 vs E16.5, $P_{adj}$:0.4018), and PPR (**D3**) recorded in E12.5, E14.5 and E16.5 CA1PNs. Left, violin plot of the three birthdate groups. Right, scatterplot of the same data against the radial position. e-IPSC amplitude (p: 0.013; $CI_{95\%}$ [0.048; 0.576]) and decay $\tau$ (p: 0.0051; $CI_{95\%}$ [–0.599; –0.097]) display a positive and negative correlation with cell location, respectively (whereas PPR does not present any correlation, p:0.3473). Overall, deeper cells are subject to larger and faster SC-associated inhibitory currents than superficial ones. PPR is higher in E12.5 CA1PNs than in E14.5 ($P_{adj}$: 0.0326; $CI_{95\%}$ [0.015.; 0.342]) and E16.5 ($P_{adj}$: 0.0177; $CI_{95\%}$ [0.083; 0.452]), without any significant difference in E14.5 vs E16.5 ($P_{adj}$:0.145). Violin plots present medians (center), interquartile ranges (bounds), minima and maxima. Color-code: E12.5 PNs: light blue, E14.5: dark blue, E16.5: magenta. The gray shaded area in scatterplots represents the thickness of the stratum pyramidale. *p < 0.05.

The online version of this article includes the following figure supplement(s) for figure 4:

**Source data 1.** Source data for evoked synaptic currents of fate-mapped CA1 PNs.

**Source data 2.** Table summarizing linear regression analyses for PSCs excluding E12.5PNs in the SR.

rheobase in respect to either position or birthdate group, despite a significant linear relationship of AP threshold and AHP with the animal age (see *Figure 6—source data 1*).

It should be noted that, although our sampling does not permit to fully separate the positional component from the embryonic origin (most recorded E14.5 cells are located in the deep portion of the stratum pyramidale and inversely for recorded E16.5 cells), none of the intrinsic electrophysiological properties measured in this experiment correlated specifically with the location in the histological location. Overall, this suggests that the embryonic birthdate is a crucial determinant of the observed cell heterogeneity, which cannot solely be explained by the radial gradient.

## Dendritic morphology of CA1PNs correlates with birthdate

We next asked whether the dendritic morphology of CA1PNs could reflect their birthdate, since pyramidal neurons with different dendritic arborizations were recently shown to distribute radially in the distal part of CA1 (*Li et al., 2017*). We used a set of neurobiotin-filled cells (E12.5 n = 30, E14.5 n = 14, E16.5 n = 19) and considered the overall dendritic arborization (*Figure 7*). We measured the distance between the cell soma and the first bifurcation of the primary apical dendrite (*Figure 7B*). Earlier-generated neurons (both E12.5- and E14.5- CA1PNs) had a significantly longer shaft than later generated neurons (*Figure 7C*, E12.5 vs E16.5, p: 0.0001; E14.5 vs E16.5, p < 0.0001). In addition, the shaft length increased linearly with the distance of the cell body from the *radiatum/pyramidale* border (r = 0.51, CI95 [0.32; 0.68], p < 0.0001, *Figure 7C*). However, we next reasoned that this correlation could be redundant, i.e. partially explained by the cell location itself, owing that a portion of the dendritic shaft includes the distance of the soma from the *radiatum/pyramidale* border (*Figure 7B*). To avoid this confounding factor, we computed as a substitute the distance between the primary apical dendrite birfucation to the *radiatum/pyramidale* border (*Figure 7C*) and observed that indeed the correlation with the soma location could no longer be found (r = 0.16, CI95 [–0.07; 0.38], p: 0.0865). However, differences related to the birthdate remained, and E14.5 CA1PNs cells were found to display a longer dendritic main branch within the *stratum radiatum* than both E12.5- and E16.5 CA1PNs (E12.5 vs E14.5, p: 0.0056; E14.5 vs E16.5, p: 0.0004, *Figure 7C*). Thus, consistent with our previous findings in CA3 (*Marissal et al., 2012*), the dendritic morphology appears to be determined by the temporal embryonic origin, at least in its basic features. Again, this correlation with birthdate is not linear since E12.5 CA1PNs were found to be more similar to E16.5 PNs than E14.5 CA1PNs, their closer peers in age.

## cFos labeling following the exploration of familiar or novel environments correlates with birthdate

The dorsal CA1 was recently shown to comprise two functional sublayers with deep CA1PNs supporting the formation of landmark-based spatial maps during a novel experience, while superficial

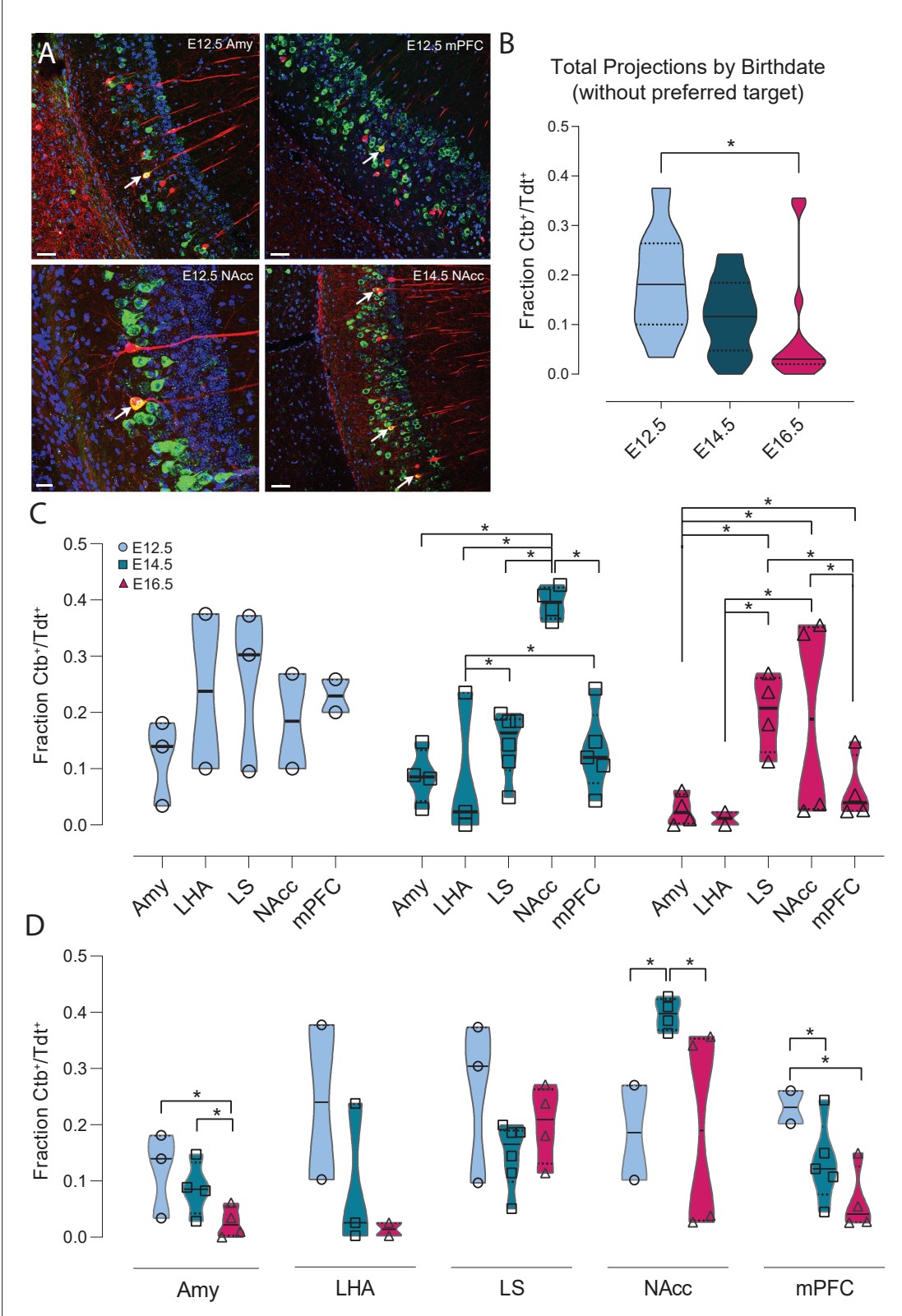

**Figure 5.** Ventral CA1 PNs with different birthdates exhibit distinct output connectivity. (**A**) Examples of cholera toxin subunit b (Ctb) retrograde labeling (green) in ventral CA1 after injection in amygdala in *Neurog2*<sup>CreER</sup>-Tdt mouse induced at E12.5 (top left), medial prefrontal cortex in E12.5 mouse (top right), Nucleus Accumbens (NAcc) in E12.5 mouse (bottom left) and NAcc in E14.5 mouse (bottom right). Colabeled Ctb⁺/Tdt⁺ cells are indicated by a white arrow. Scalebars: top left, top right and bottom right, 50 μm; bottom left, 20 μm. (**B**) Quantification of the total Ctb⁺/Tdt⁺ cell fraction for the

*Figure 5 continued on next page*

Figure 5 continued

three birthdates, excluding the preferred region for each (E12.5 = lateral septum (LS), E14.5 = NAcc, E16.5 = LS). Overall, significantly more $Ctb^+/Tdt^+$ were found in E12.5 PNs than E16.5 ($P_{adj}$: 0.0129, $CI_{95\%}$ [0.013; 0.19]) while no significant difference was observed when comparing E12.5 with E14.5 ($P_{adj}$: 0.070) or E16.5 PNs ($P_{adj}$: 0.146). (**C**) Fraction of $Ctb^+/Tdt^+$ cells in E12.5, E14.5 and E16.5 PNs by target region. Note how E12.5 neurons project more homogeneously to all structures probed, while marked inter-regional differences appear among E14.5 and E16.5 neurons. (**D**) Fraction of $Ctb^+/Tdt^+$ cells in amygdala (Amy), lateral hypothalamic area (LHA), lateral septum (LS), nucleus accumbens (NAcc), and medial prefrontal cortex (mPFC) by time of neurogenesis. E12.5 PNs project more prominently than other birthdate groups to Amy and mPFC, while NAcc is preferentially targeted by E14.5.

The online version of this article includes the following figure supplement(s) for figure 5:

**Source data 1.** Table summarizing fate-mapped $Ctb^+$-$Tdt^+$ PN co-labeling data.

**Figure supplement 1.** Representative sections of injections sites.

**Figure supplement 2.** Cell location of fate-mapped $Ctb^+/Tdt^+$ PNs in ventral CA1.

---

CA1PNs were more active in cue-poor conditions and likely to convey self-referenced information (*Danielson et al., 2016*; *Fattahi et al., 2018*; *Geiller et al., 2017*; *Sharif et al., 2020*). We asked whether similar differences could be observed in the ventral CA1 and whether they could reflect developmental origin, given the different connectivity schemes and excitability across CA1PNs with different birthdates. To this aim, we tested the activation of birthdated ventral pyramidal neurons during the exploration of an environment, by the means of cFos immunoreactivity, given the poor accessibility of the ventral hippocampus to imaging and the sparsity of pioneer CA1PNs. cFos is an immediate early gene whose expression does not simply reflect previous activity in labelled neurons but also the induction of activity-related plasticity (*West et al., 2002*).

$Neurog2^{CreER}$-Tdt mice (n = 25) induced at E12.5, E14.5, and E16.5 were divided into three groups (*Figure 8A*): one explored a cue-enriched arena during 20 min for 3 consecutive days, the familiar group (FAM, n = 8 mice), whereas another group was exposed to the same environment during only one session on the third day, the novel group (NOV, n = 10). A control group was handled by the experimenter but did not explore any arena, the home cage group (HC, n = 7). As expected, FAM mice decreasingly explored the arena throughout the three consecutive sessions (repeated measures one-way ANOVA, test for trend: p < 0.01, *Figure 8A*). Coherently, the distance run by NOV mice was significantly higher than that of the second and third FAM sessions (one-way ANOVA, NOV vs FAM2, p < 0.001; NOV vs FAM3, p < 0.001), but not of the first session (FAM1). As expected, the overall proportion of $cFos^+/DAPI$ neurons was significantly higher in NOV (10.76 % ± 3.42) and FAM (9.40 % ± 1.31) than the HC condition (5.44 % ± 2.45), further confirming that cFos expression in hippocampal neurons is up-regulated upon exploratory novelty (*Figure 8B*). Next, we reasoned that thigmotaxis could provide a proxy of anxiety-like behavioral state in our experimental setting. We therefore measured the time spent in the center (more anxiogenic) of the rectangular arena. We found no linear correlation between cFos expression rate and the percentage of time spent exploring the central zone (see *Figure 8—source data 1*), suggesting that our protocol is not well suited for probing anxiety-like behavior through the analysis of cFos activation.

To study how cFos activation may vary according to a neuron's embryonic origin, we first computed the proportion of $cFos^+/Tdt^+$ cells by birthdate group, regardless of the condition (*Figure 8C*). Doing so, we observed that E16.5 mice displayed significantly lower proportions of co-labeled cells than both E12.5 (95% Confidence interval or $CI_{95\%}$ [0.0097; 0.0483], p < 0.01) and E14.5 ($CI_{95\%}$ [0.0137; 0.0459], p < 0.001). Hence, late-born E16.5 CA1PNs are less prone to express cFos, as expected from the previously reported global difference in overall recruitment between deep and superficial CA1PNs (*Mizuseki et al., 2011*). Using the same approach as for retrograde tracing analysis, we examined the $cFos^+/Tdt^+$ fraction per condition per group. We found that pioneer E12.5 CA1PNs were highly likely to express cFos following habituation (E12.5-FAM), as compared with the home cage within the same birthdate group (E12.5-FAM vs E12.5-HC, p < 0.001; E12.5-FAM vs E12.5-NOV, p: 0.066) and across birthdates for the same FAM condition (E12.5-FAM vs E14.5-FAM, p < 0.01; E12.5-FAM vs E16.5-FAM, p < 0.05, *Figure 8C*). Although more than twice less than for E12.5 CA1PNs, E16.5 CA1PNs were also significantly more likely to express cFos in FAM than HC conditions. Overall, these cells expressed cFos at very low rate, especially in the HC condition where co-labeling was significantly lower than observed in E14.5 mice (E16.5-HC vs E14.5-HC, p < 0.01). Importantly, in the NOV condition, earlier-generated cells showed more $cFos^+$ activation than later born ones, although this only appeared significant for E14.5 CA1PNs (E14.5-NOV vs E16.5-NOV, p < 0.05; E14.5-NOV vs E12.5-NOV, p:

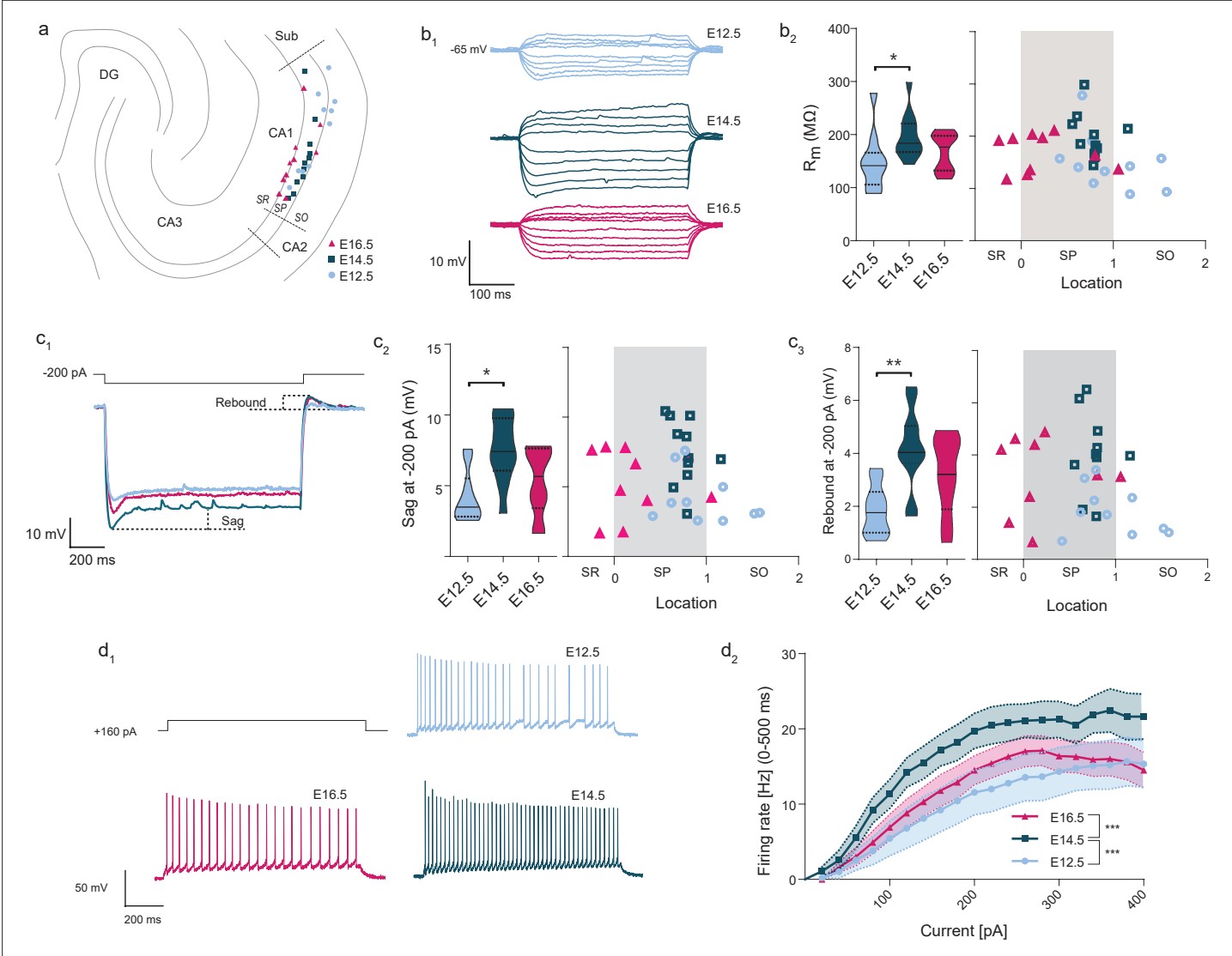

**Figure 6.** Intrinsic excitability varies according to embryonic birthdate. (**A**) Anatomical location of neurobiotin-filled CA1PNs recorded from acute horizontal slices in current-clamp experiments. DG: dentate gyrus, Sub: subiculum, SR: stratum radiatum, SP: stratum pyramidale, SO: stratum oriens. (**B1**) Representative membrane responses to a series of hyper- and depolarizing current steps recorded from fate-mapped CA1PNs. Note the larger deflection of membrane potential in the E14.5 neuron. (**B2**) Membrane input resistance (Rm) of fate-mapped CA1PNs. Left, violin plot of the three birthdate groups. Right, scatterplot of the same data against the radial position. Rm is higher in E14.5 than E12.5 cells ($P_{adj}$: 0.025; $CI_{95\%}$ [–89.71; –11.15]; E12.5 vs E16.5, E16.5 vs E14.5, $P_{adj}$:0.348). No linear correlation was found with cell location (p:0.139). (**C1**) Sag potential response and rebound of fate-mapped CA1PNs, following –200 pA current injections. In this example, E14.5 CA1PNs display a greater sag response than E12.5 and E16.5 PNs. (**C2**) Sag response. Left, violin plot of the three birthdate groups. Right, scatterplot of the same data against the radial position. Sag response is significantly higher in E14.5 than E12.5 cells ($P_{adj}$: 0.0026; $CI_{95\%}$ [1.74; 6.05]; E12.5 vs E16.5, E16.5 vs E14.5, $P_{adj}$:0.128). No linear correlation was found with cell location (p:0.27). (**C3**) Rebound potential. Left, violin plot of the three birthdate groups. Right, scatterplot of the same data against the radial position. Sag response is significantly higher in E14.5 than E12.5 cells ($P_{adj}$: P:0.0063; $CI_{95\%}$ [–3.82;–1.29]; E12.5 vs E16.5, $P_{adj}$:0.054; E16.5 vs E14.5, $P_{adj}$:0.223). No linear correlation was found with cell location (p:0.139). (**D1**) Examples of firing responses to a depolarizing current step recorded from fate-mapped CA1PNs. Note that the number of action potential fired by E16.5 and E12.5 PNs is lower than E14.5 PNs. (**D2**) Input-output curves of fate-mapped CA1PNs. E14.5 PNs have a higher firing rate than E12.5 PNs ($P_{adj}$ < 0.0001; $CI_{95\%}$ [5.21; 11.14]) and E16 ($P_{adj}$ < 0.0001; $CI_{95\%}$ [–9.91; –4.13]), suggesting that they are more excitable (E16.5 vs. E12.5, $P_{adj}$:0.645). Effect of current injection $F_{(2, 540)}$ = 25.65, p < 0.0001; effect of birthdate $F_{(19, 540)}$ = 10.71, p < 0.0001; interaction $F_{(38, 540)}$ = 0.2355, p > 0.9999, ordinary two-way ANOVA with Tukey's post hoc test. Data are represented as means ± standard errors of the means. Violin plots present medians (center), interquartile ranges (bounds), minima and maxima. Color-code: E12.5: light blue, E14.5: dark blue, E16.5: magenta. The gray shaded area in scatterplots represents the thickness of the stratum pyramidale. *p < 0.05; **p < 0.01; ***p < 0.001.

The online version of this article includes the following figure supplement(s) for figure 6:

**Source data 1.** Source data for intrinsic membrane properties of fate-mapped CA1 PNs.

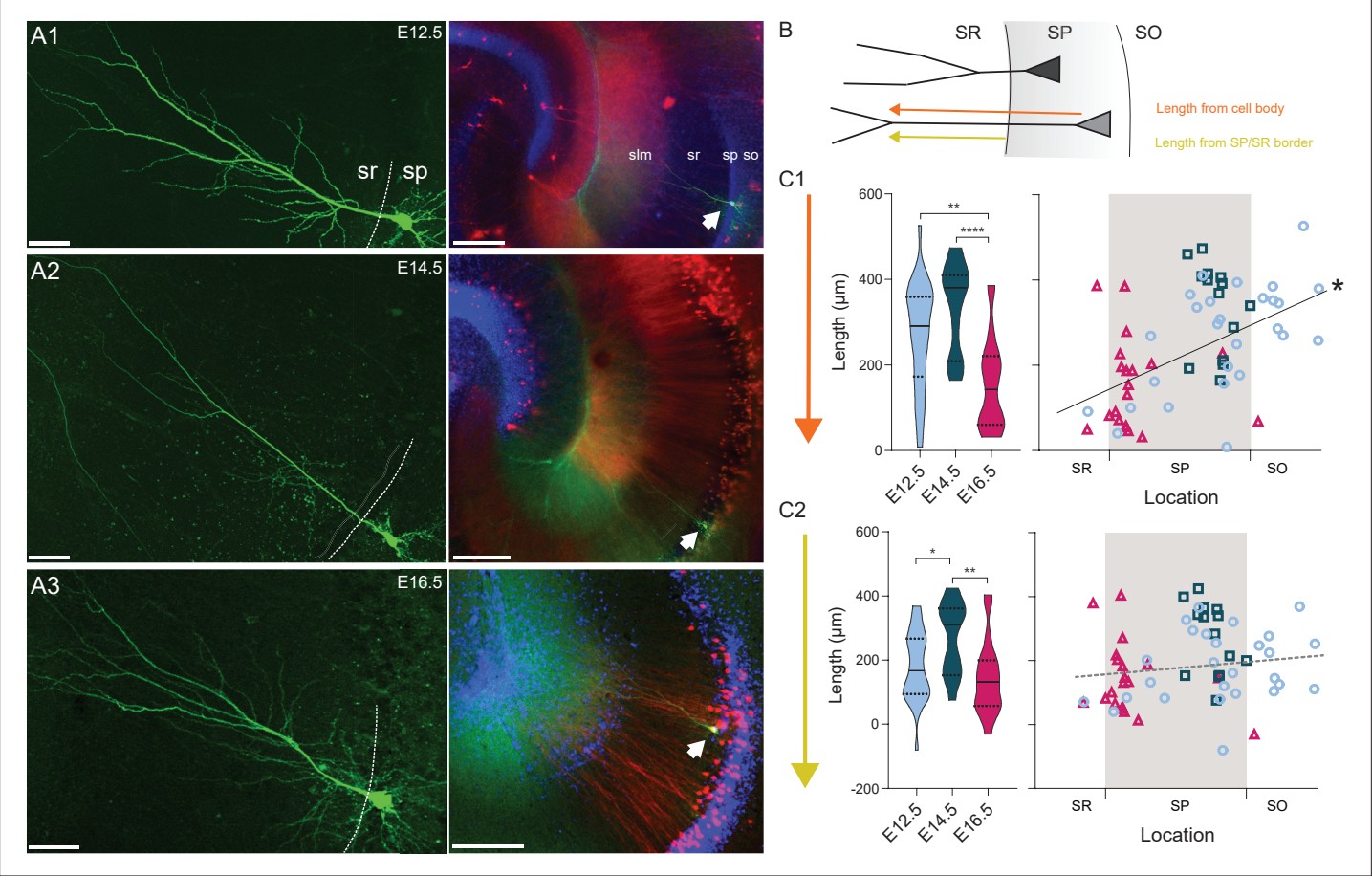

**Figure 7.** Embryonic origin is a determinant of dendritic morphology. (**A1-3**) On the left, representative examples of CA1 PNs dendritic arborization in neurobiotin-filled cells. The dashed line is drawn at the border between stratum pyramidale (SP) and stratum radiatum (SR). On the right, the same cell as on the left is shown at lower magnification and indicated by a white arrow. Red and blue channels represent Tdtomato and DAPI, respectively. (**A1**) E12.5 PN, (**A2**) E14.5 PN, (**A3**) E16.5 PN. Note that the main dendritic branch bifurcates more distally in E14.5 cells. Scalebars: left, 50 μm; right, 200 μm. so: stratum oriens; sp: stratum pyramidale; sr: stratum radiatum; slm: stratum lacunosum-moleculare. (**B**) Scheme illustrating the primary dendrite length measured from soma or from the border between SP and SR to the first major dendritic bifurcation. SO: stratum oriens. (**C1**) Primary dendrite length from soma to bifurcation in fate-mapped CA1PNs. Left, violin plot of the three birthdate groups. Right, scatterplot of the same data against the radial position. E16.5PNs display a reduced dendrite length, in respect to E12.5 (p: 0.0001; CI$_{95\%}$ [37.74; 190.36]) and E14.5 PNs (p < 0.0001; CI$_{95\%}$ [92.07; 262.82]; E12.5 vs E14.5, p: 0.031. Sidak's alpha for significance: 0.0169). The dendrite length from soma is markedly correlated with the soma location (p < 0.0001; CI$_{95\%}$ [0.318; 0.683]), likely due to the contribution of the dendritic segment within the SP. (**C2**) Primary dendrite length measured between the SP/SR border and the dendrite bifurcation in fate-mapped CA1PNs. Left, violin plot of the three birthdate groups. Right, scatterplot of the same data against the radial position. The dendritic length is larger in E14.5 than E12.5 PNs (P$_{adj}$: 0.0056; CI$_{95\%}$ [−172.99; −6.14]) and E16 cells (P$_{adj}$: 0.0004; CI$_{95\%}$ [36.48; 217.12], E12.5 vs E16.5, p: 0.116. Sidak's alpha for significance: 0.0169). No linear correlation was found with the cell location (P:0.109). Violin plots present medians (center), interquartile ranges (bounds), minima and maxima. Color-code: E12.5: light blue, E14.5: dark blue, E16.5: magenta. The gray shaded area in scatterplots represents the thickness of the stratum pyramidale. *p < 0.05. **p < 0.01. ****p < 0.0001.

The online version of this article includes the following figure supplement(s) for figure 7:

**Source data 1.** Source data for morphometric properties of fate-mapped CA1 PNs.

0.165). Taken together, these results are in line with previous research, in that deep CA1PNs are more likely to express IEGs than superficial ones upon exposure to a novel environment (*Kitanishi et al., 2009*). Yet, pioneer E12.5 CA1PNs form a distinct subpopulation more specifically expressing cFos after familiarization to the environment. Hence, these last results indicate a different involvement in novelty detection of pyramidal neurons in the ventral CA1 according to their developmental origin.

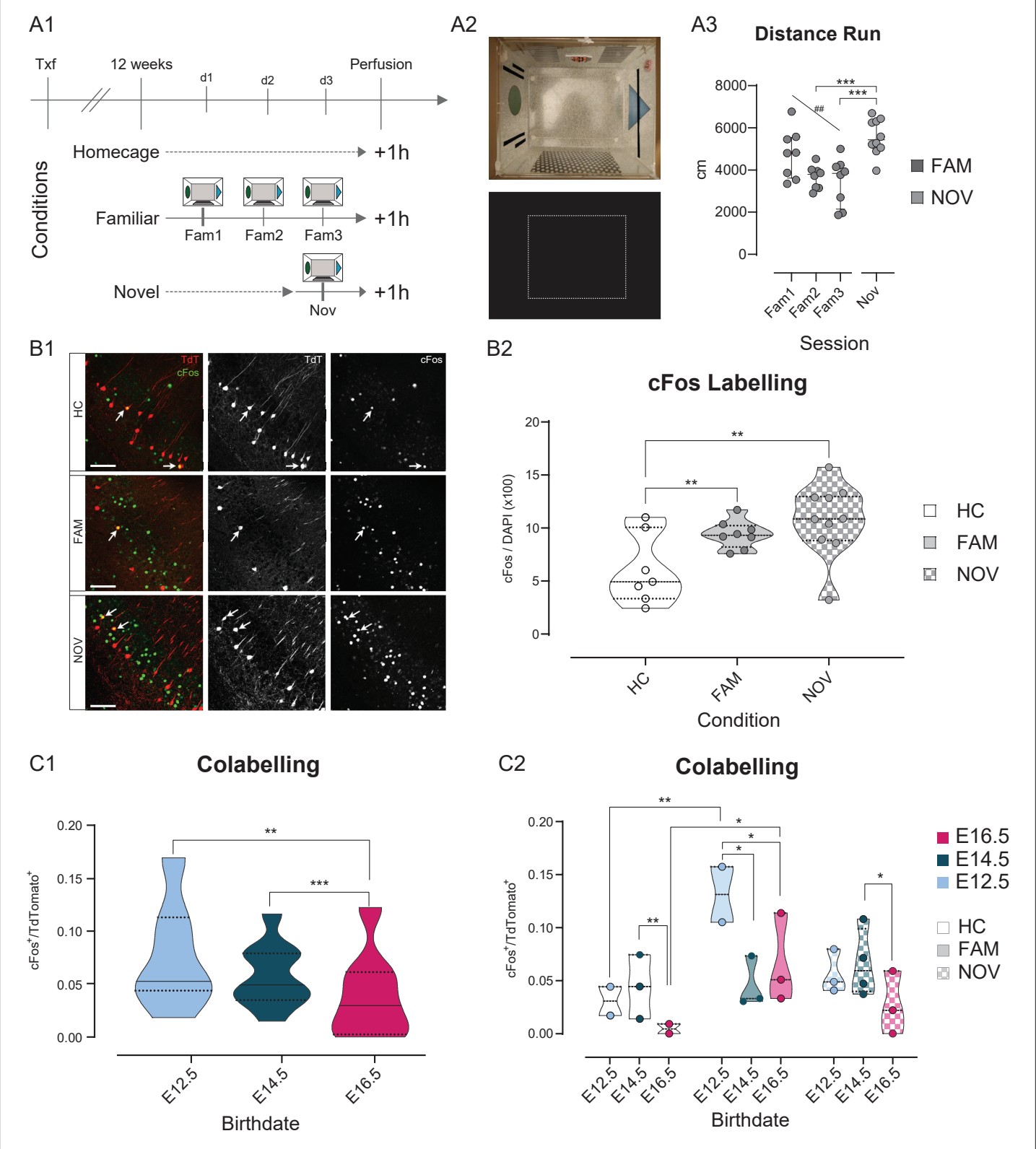

**Figure 8.** Fate-mapped CA1PNs differ in cFos expression following exploratory behavior. (**A1-3**) Experimental paradigm and validation for detecting cFos expression upon exploration of an arena. (**A1**) Schematic representation of the behavioral conditions, with 3 cohorts of tamoxifen-induced *Neurog2^{CreER}*-Tdt mice: homecage (HC), no exploration, familiar (FAM), repeated exploration (20') on three consecutive days, and novel (NOV), only one exploration (20'). (**A2**) Top, view from above of the exploration box containing visual, tactile, and olfactory (butanal) cues. In addition, a white noise was

*Figure 8 continued*

played in the experimental room. Bottom, representative occupancy heat map. (**A3**) Quantification of the distance run by animals during exploration of the arena. Upon repeated exposure (Fam1-3), mice explored progressively less the arena (RM one-way ANOVA, main effect = F(1,14): 11.52, p: 0.0044, Slope:–660.1 ± 194.5, CI$_{95\%}$ of slope [–242.95;–1077.2]), suggesting that novelty decreases over sessions. Also, NOV did not differ significantly from Fam1 (P$_{adj}$: 0.26, CI$_{95\%}$ [–323.80; 1921.7]), where mice explored for the first time, and was higher than Fam3 (P$_{adj}$: 0.0006, CI$_{95\%}$ [973.4; 3264.8]), corresponding to the last exploration (one-way ANOVA, main effect = F(3,30): 9.6745, p:0.0001). (**B1-2**) The expression of cFos in ventral CA1 is higher upon exploration of a novel environment, than after repeated exposures. (**B1**) Immunohistochemistry anti-cFos, performed on three E14.5 tamoxifen-induced mice. Top, HC; middle, FAM; bottom, NOV condition. Each shows the merged Tdt, cFos and DAPI image (left), Tdt (center), cFos (right). Note the increase in cFos signal from HC to FAM and NOV. Insert: magnification on two cells, one strongly cFos$^+$, the other co-expressing Tdt and cFos. Scalebar: 100 µm. (**B2**) Fraction of cFos$^+$ cells per animal. In both FAM (P$_{adj}$: 0.0006, CI$_{95\%}$ [–5.78; –1.99]) and NOV (P$_{adj}$: 0.0006; CI$_{95\%}$ [–7.86; –2.54]) animals, cFos immunoreactivity is increased compared to HC (Fam vs Nov, P$_{adj}$:0.111). (**C1-2**) Quantification of the co-expression of cFos and Tdt in fate-mapped CA1PNs. (**C1**) Fraction of cFos$^+$/Tdt$^+$ cells by birthdate group, regardless of the condition. E16 cells display fewer co-labelled cells than E12 (P$_{adj}$: 0.0022, CI$_{95\%}$ [0.0097; 0.0483]) and E14 (P$_{adj}$: 0.0009, CI$_{95\%}$ [0.0137; 0.0459]; E12.5 vs E14.5, P$_{adj}$:0.464). (**C2**) Fraction of cFos$^+$/Tdt$^+$ cells by birthdate and per condition. Each animal is represented as a datapoint indicating the fraction of cFos$^+$/Tdt$^+$ cells. Note that within the FAM condition, cFos$^+$/Tdt$^+$ cells are more abundant in E12-FAM than E14-FAM (P$_{adj}$: 0.008, CI$_{95\%}$ [0.0284; 0.135]) and E16-FAM (P$_{adj}$: 0.0325, CI$_{95\%}$ [0.0236;0.136]). Within E12 birthdate group as well, E12-FAM fraction of co-labeled cells is higher than E12-HC (P$_{adj}$ < 0.0001, CI$_{95\%}$ [–0.151; –0.049]), but not than E12-NOV (P$_{adj}$ < 0.066, CI$_{95\%}$ [0.0149; 0.123]). In addition, E14-NOV is greater than E16-NOV (P$_{adj}$ < 0.015, CI$_{95\%}$ [0.0179; 0.0797]). Violin plots present medians (center), interquartile ranges (bounds), minima and maxima. Color-code: E12: light blue, E14: dark blue, E16: magenta. *p < 0.05. ##,**p < 0.01. ***p < 0.001.

The online version of this article includes the following source data and figure supplement(s) for figure 8:

**Source data 1.** Source data of mouse locomotory activity for each behavioral session.

**Source data 2.** Table summarizing DAPI, cFos$^+$ and cFos$^+$-Tdt$^+$ labeling data following exploration.

**Figure supplement 1.** Cell location of fate-mapped cFos$^+$/Tdt$^+$ PNs in ventral CA1.

## Discussion

In this study, we have employed a manifold description of CA1PNs with different birthdates in the adult hippocampus to study the origin of diversity at single-cell level. We have found that the now well-established subdivision of the CA1 pyramidal layer into two distinct sublayers, a deep and super-ficial one, masks a greater heterogeneity, whose logic can be partly exposed when considering the temporal origin of individual neurons. This heterogeneity encompasses several morpho-physiological properties and translates in the propensity of cells to be recruited in familiar or novel environments. Furthermore, we find that pioneer CA1PNs generated at the earliest stages of glutamatergic cell neurogenesis are prone to be recruited in familiar environments, forming a subpopulation with distinct anatomical distribution, morphophysiological features and wiring.

### Birthdate rather than birth order is a better predictor of diversity

Several studies converge in suggesting that the CA1 pyramidal layer comprises two subtypes of CA1PNs based on their radial positioning. We would like to propose that the diversity of CA1PNs is even better predicted by birthdate than position. More specifically, we found that the following metrics were better reflecting birthdate than radial positioning based on the lack of correlation with soma position and the existence of a significant correlation with birthdate: (1) the length of primary apical dendrite in the *stratum radiatum*; (2) the input resistance, sag and rebound responses, and firing rate upon depolarizing current injections; (3) the E/I ratio; (4) the short-term facilitation of evoked synaptic inhibitory currents. The two main metrics that linearly correlate with layering reflect the local microcircuit integration of CA1PNs as they are the frequency of sEPSCs and the evoked IPSC decay.

It was previously reported that both ectopic CA1PNs with a soma located in the *stratum oriens* received a higher rate of sEPSCs than cells located within the *stratum pyramidale* (**Cossart et al., 2000**). Similarly, we find that CA1PNs located closer to the *stratum oriens* receive sEPSCs at a higher rate. This could reveal either a slicing artefact or the contribution of glutamatergic inputs from the CA1 axonal collaterals arborizing in that area. The faster kinetics of evoked IPSCs may reflect the increased contribution of parvalbumin basket cell inputs to evoked IPSCs as cells move towards the *oriens*. Indeed, the alpha1-containing postsynaptic GABAa-Rs contributing to PV synapses were shown to display faster kinetics than those formed by CCK basket cells (**Thomson et al., 2000**). In line with studies indicating specified microcircuitry among deep versus superficial principal cells and PV basket cells, we have found that the PV staining was denser towards the *stratum oriens*. However, within the deep CA1PNs, cells born at E12.5 were exceptionally less decorated with putative PV contacts

than deep CA1PNs born 2 days later. Besides, like E16.5 CA1PNs, E12.5 CA1PNs were more likely to express calbindin than E14.5 CA1PNs, even when there soma was located in the deep layer. Overall, we observed that E12.5 cells were more similar to E16.5 CA1PNs regarding their lower intrinsic excitability, dendritic morphology, lower PV coverage, higher calbindin expression and higher E/I ratio. As E16.5 CA1PNs, they can even be found in the superficial layer and even in the *stratum radiatum*. Altogether, one could foresee that the main source of activation for both E12.5 and E16.5 CA1PNs would be through synaptic glutamatergic excitation. However, it is likely that the main glutamatergic inputs onto E16.5 and E12.5 CA1PNs differ. While the main excitatory drive onto E16.5 CA1PNs, given their location likely originates in CA3, future studies are needed to determine whether E12.5 cells receive specific inputs. Such studies remain at the moment technically challenging due to the poor accessibility of E12.5 cells to genetic manipulation using our fate-mapping strategy.

These cells also differ in their extrahippocampal outputs. Accordingly, these cells could also be discriminated based on their cFos expression following spatial exploration with E12.5, like E14.5 CA1PNs being more likely to express cFos during random exploration than E16.5 CA1PNs. Future studies are needed to determine the conditions favoring cFos expression in E16.5 CA1PNs. These may require analyzing these cells in conditions of elevated anxiety, social interactions or goal-directed behaviors (*Ciocchi et al., 2015*; *Jimenez et al., 2018*; *Okuyama et al., 2016*). If E12.5 CA1PNs and E14.5 CA1PNs displayed comparable cFos expression levels, as expected from previous work (*Mizuseki et al., 2011*), the former signals familiarity and the latter novelty, suggesting that the mechanisms supporting hippocampal representation of novelty may require cells with a higher intrinsic neuronal excitability (*Epsztein et al., 2011*). In line with previous results, our data show an increase in cFos expression during novel exploration as compared to the control condition. However, we fail to show a significant difference between NOV and FAM. The disparity between our findings and those from other studies (*VanElzakker et al., 2008*) might owe to the experimental design (notably the number of exposures to a given environment), explaining different degrees of familiarity reached by the animals.

Most of our results are in agreement with previous reports with the exception of the correlation between radial position and sag, membrane potential or input resistance. Only E12.5 CA1PNs could be distinguished by their lower sag amplitude value and input resistance. This apparent discrepancy may have several explanations. First, our experiments were not performed in the presence of blockers for other postsynaptic membrane currents and of antagonists for all metabotropic and ionotropic GABA and glutamate receptors (*Jarsky et al., 2008*; *Lee et al., 2014*; *Maroso et al., 2016*; *Masurkar et al., 2017*). Second, we focused on ventral and not dorsal hippocampus (*Maroso et al., 2016*), and cells were sampled evenly across the transverse axis, while radial correlations could only be revealed at fixed positions along the proximo-distal axis (*Masurkar et al., 2017*). We were also surprised not to observe any bursting cell (*Jarsky et al., 2008*; *Graves et al., 2012*), but again these are mainly present closer to the subiculum and were reported in slices from juvenile (P15-17 *Jarsky et al., 2008*; P25-28 *Graves et al., 2012*) rats using a gluconate-containing intracellular solution (*Kaczorowski et al., 2007*). Last, while we could observe a linear correlation between soma location and dendritic morphology as reported previously (*Graves et al., 2012*; *Jarsky et al., 2008*; *Li et al., 2017*; *Lee et al., 2014*), this was no longer observed when considering only the dendritic portion within the *stratum radiatum*. When computing this metric, E14.5 CA1PNs could be distinguished from E12.5 and E16.5 CA1PNs by their longer primary apical segment. Also, one needs to consider one major limitation of the fate-mapping approach employed here when interpreting the results of the present study. Indeed, our labeling is based on the Cre-dependent and tamoxifen-induced expression of a fluorescent reported protein in *Neurog2* expressing progenitors. This is a stochastic process, and a variable and sparse subset of progenitors gets labeled. We are only describing small numbers and the most distinctive features are more likely to show up as significantly different whereas other metrics may require denser sampling.

Altogether, our results globally indicate that the order of neurogenesis does not imprint a continuous gradient in the specification of cell laminar positioning and diversity. Instead, these are predetermined by the specific temporal origin of individual cells. Interestingly, the notion that early specification has a stronger importance than final layering in the determination of CA1PNs intrinsic properties is supported by recently observations with a transgenic mouse model of lissencephaly where CA1 lamination is impaired while cell identity is relatively spared (*D'Amour et al., 2020*). Therefore, the

recently uncovered system of parallel information processing in CA1, with two information streams segregated through distinct inhibitory domains (*Soltesz and Losonczy, 2018*), may need to consider this additional level of complexity.

The mechanisms by which diversity among CA1PNs may be temporally-regulated during neurogenesis remain to be elucidated and would require specific genetic manipulation of PNs with different birthdates. Increasing evidence indicates that glutamatergic neurons are issued from the same multipotent progenitors and that fate distinctions are mostly temporally controlled (*Jabaudon, 2017*) (but see *Franco et al., 2012*). Interestingly, progenitor potential was shown to be progressively, temporally restricted, with early cortical progenitors being multipotent in comparison to later ones (*Lodato et al., 2015*). In combination with genetic predetermination, single-cells display tightly orchestrated sequences of spontaneous activity patterns (*Allene et al., 2012*), which in turn may contribute to the maturation of physiological specificity. Therefore, cells with similar birthdates will follow similar activity development schedules. Interestingly, both Ih and Rm, two parameters that segregate between CA1PNs with different birthdates, are developmentally regulated, both progressively decreasing as cells mature (*Dougherty, 2019*). As such, they are ideal proxys of neuronal maturation stage. Given that the earlier born CA1PNs can be distinguished by lower values of Ih and Rm, one could propose that these cells could maintain into adulthood the advance in maturation originating from their early birthdate.

## Pioneer cells are a different population, possible roles?

This study shows that neurons born at the earliest phases of CA1PNs neurogenesis display distinct somatic distribution, connectivity, morpho-physiological properties as well cFos expression patterns. This distinguishes them from slightly later born neurons, the E14.5 CA1PNs that sit mainly in the deep sublayer of the *stratum pyramidale*. We argue that these cells form a distinct subpopulation. The first striking property of E12.5 CA1PNs is their scattered distribution. They can be found at ectopic positions in the *stratum oriens* as well as, more rarely, within the *stratum radiatum*, suggesting that these cells may also comprise the previously described radiatum 'giant cells', with privileged CA2 input and output to the olfactory bulb (*Gulyás et al., 1998*; *Nasrallah et al., 2019*). In fact, cumulative distribution plots indicate that roughly half of the E12.5 CA1PNs in the ventral (and dorsal) hippocampus are found within less than 100 μm distance from the *stratum radiatum* while the other half distributes closer to *stratum oriens* more than 100 μm away from that border. It is unlikely that such observation results from unspecific labeling from our method. First, the population of E12.5 CA1PNs labeled using this method shares many characteristics despite this layer dispersion. Second, this unique arrangement was also recently observed for early born granule cells (*Save et al., 2019*) and dispersion of early generated glutamatergic cell cohorts was previously suggested using other methods including autoradiographic (*Caviness, 1973*) and retrovirus labeling (*Mathews et al., 2010*). Somehow similarly, even if preferentially located in the deep *stratum pyramidale*, early born CA3 pyramidal cells could be found anywhere from *stratum oriens* to *lucidum* (*Marissal et al., 2012*). It is therefore possible that pioneer cohorts of PNs would display a different migration mode, resulting from the low cellular density when entering the hippocampus in addition to other more specific mechanisms, like an absence of radial glia bending (*Xu et al., 2014*), that remain to be specifically studied. Regardless, despite this almost even positioning, we observed that E12.5 CA1PNs displayed specific input connectivity schemes, including a lower number of putative PV contacts associated with a higher E/I ratio and a facilitation of the evoked IPSC amplitude. We hypothesize that these three measurements may reveal the same feature, namely the lower somatic inhibitory coverage of E12.5 CA1PNs. Indeed, facilitating inhibitory responses were found to originate from dendrite-targeting interneurons while perisomatic cells would tend to generate depressing inhibitory inputs (*Pouille and Scanziani, 2004*). This lower PV innervation is comparable with E16.5 CA1PNs and contrasts with their early-born peers, the E14.5 CA1PNs, which display higher perisomatic PV staining. Whether these CA1PNs are contacted by specific glutamatergic inputs stays an open question that remains an experimental challenge with our fate-mapping strategy. According to the temporal matching rule observed in the hippocampus (*Deguchi et al., 2011*), and the fact that some of them are found in the *stratum radiatum* and *oriens*, one may expect these cells to receive preferential innervation from CA2, the earliest generated portion of the hippocampal circuit (*Caviness, 1973*). The significantly higher E/I ratio received by these cells suggests that they may be preferentially recruited by synaptic excitation,

since they are otherwise less intrinsically excitable (lower Rm, low firing rate in response to long depolarizing current injections, lower sag). Interestingly, the same lower excitability (across similar metrics), combined with lower levels of PV inputs was observed in early born CA1 GABAergic neurons and DG granule cells (*Bocchio et al., 2020*; *Save et al., 2019*; *Gupta et al., 2012*), again indicating similar fates in the adult across different subtypes of pioneer neurons.

Ventral E12.5 CA1PNs are also remarkable in terms of output since they globally sent more projections to the target regions studied here and did not display any preferred projection pattern in contrast to their later born peers. Of note they were significantly more likely to target the Amy and mPFC than both E14.5 and E16.5 cells. These results would suggest that E12.5 CA1PNs may form a sparse population with multiple projections, similar to the triple projection ventral CA1 cells reported by Ciocchi and colleagues (*Ciocchi et al., 2015*). However, the low yield of our fate-mapping pioneer cells combined with the one of multiple retrograde labeling precluded addressing this possibility. Interestingly, such triple projection ventral CA1 cells targeting the mPFC, Amy and NAcc were shown to be highly active and in particular during SWRs (*Ciocchi et al., 2015*), which would be in agreement with the fact that E12CA1 PNs receive fewer putative PV contacts. Unfortunately, the sparsity of our labeling method currently prevents opto-tagging E12.5 CA1PNs as well as testing the hypothesis that they may be comprised of multiple projection neurons.

## Conclusion

We have uncovered a novel population of pioneer cells adding to the diversity of CA1 glutamatergic neurons. Based on their specific properties, we would like to propose a general role for ventral pioneer CA1PNs in the consolidation or retrieval of recent experience. Pioneer CA1PNs appear as a relatively sparse subpopulation, similar to the small minority of fast firing 'rigid' neurons (*Mizuseki and Buzsáki, 2013*; *Grosmark and Buzsáki, 2016*), however their functional role may be amplified by their various outputs. Future work examining their recruitment in SWRs (*Ciocchi et al., 2015*), their possible CA2 inputs (*Kohara et al., 2014*; *Nasrallah et al., 2019*; *Valero et al., 2015*) and their likely multiple projection targets will certainly inform about this possibility. In any case, the present results indicate that the radial subdivision of the CA1 pyramidal layer into two functional sublayers needs to be revisited by also considering the time of neurogenesis. This initial blueprint may render these cells more resilient to diseases resulting in heterotopias and mislayering, as recently reported (*D'Amour et al., 2020*).

## Materials and methods

**Key resources table**

| Reagent type (species) or resource | Designation | Source or reference | Identifiers | Additional information |
|---|---|---|---|---|
| Strain, strain background (*Mus musculus*) | RjOrl:SWISS | Mouse Genome Informatics | MGI:6430821 | |
| Gene (*Mus musculus*) | Neurog2tm1(cre/Esr1*)And | Mouse Genome Informatics | MGI:2652037 | |
| Gene (*Mus musculus*) | Gt(ROSA)26Sortm14(CAG-tdTomato)Hze | Mouse Genome Informatics | J:155,793 | |
| Antibody | Anti-parvalbumin (goat polyclonal) | Swant | RRID:AB_10000345 | (1:1000) |
| Antibody | Anti-cholera toxin beta subunit (rabbit polyclonal) | Invitrogren | RRID:AB_779810 | (1:1000) |
| Antibody | Anti-cFos (rabbit polyclonal) | Abcam | RRID:AB_190289 | (1:5000) |
| Antibody | Anti-CBD28k (rabbit polyclonal) | Swant | RRID:AB_10000340 | (1:1000) |
| Antibody | AlexaFluor647-conjugate Anti-rabbit (donkey polyclonal) | Jackson Immunoresearch | RRID:AB_2492288 | (1:500) |
| Antibody | AlexaFluor647-conjugate Anti-rabbit (goat polyclonal) | Jackson Immunoresearch | RRID:AB_2338072 | (1:500) |

*Continued on next page*

*Continued*

| Reagent type (species) or resource | Designation | Source or reference | Identifiers | Additional information |
|---|---|---|---|---|
| Antibody | AlexaFluor488-conjugate Anti-goat (donkey polyclonal) | Jackson Immunoresearch | RRID:AB_2336933 | (1:500) |
| Chemical compound, drug | AlexaFluor647-conjugate Cholera Toxin subunit b | ThermoFisher | Catalog ID:C34778 | (0.1%) |
| Chemical compound, drug | Neurobiotin | Vectorlabs | Catalog ID:SP-112 | (0.5%) |
| Chemical compound, drug | Streptavidin-AlexaFluor488 | Thermofisher | Catalog ID:S11223 | (1:1000) |
| Software, algorithm | Patchmaster | Heka | RRID:SCR_000034 | |
| Software, algorithm | MATLAB | MathWorks | RRID:SCR_001622 | |
| Software, algorithm | GraphPad Prism | GraphPad | RRID:SCR_015807 | |
| Software, algorithm | MiniAnalysis | Synaptosoft | RRID:SCR_002184 | |
| Software, algorithm | CellProfiler | CellProfiler | RRID:SCR_007358 | |
| Software, algorithm | ZEN | Zeiss | RRID:SCR_013672 | |
| Software, algorithm | EthoVision XT | Noldus | RRID:SCR_000441 | |
| Software, algorithm | Fiji | Fiji | RRID:SCR_002285 | |
| Other | DAPI stain | Vector Laboratories | Catalog ID:H-1200 | Mounting medium |

## Animals

All protocols were performed under the guidelines of the French National Ethics Committee for Sciences and Health report on 'Ethical Principles for Animal Experimentation' in agreement with the European Community Directive 86/609/EEC under agreement #28.506. All efforts were made to minimize pain and suffering and to reduce the number of animals used. Animals were maintained with unrestricted access to food and water on a 12 hour light cycle, and experiments were conducted during the light portion of the day. To mark glutamatergic neurons generated during different times of embryogenesis, we use the technique of inducible genetic fate-mapping (see *Marissal et al., 2012*; *Save et al., 2019*). In brief, double heterozygous $Neurog2^{CreER+/-}/Rosa26< lsl-tdTomato^{+/-} ±$ (referred as $Neurog2^{CreER}$-Tdt in the text for simplicity) were obtained by crossing $Neurog2^{CreER+/-}/Rosa26< lsl-tdTomato^{+/+}$ male mice with 7–8 week old wild-type Swiss females (C.E Janvier, France). To induce CreER activity during embryogenesis, tamoxifen was delivered with a silicon-protected needle (Fine Science Tools) by gavaging (force-feeding) pregnant mothers at embryonic days E12.5, E14.5 and E16.5. We used 2 mg of tamoxifen solution (Sigma, St. Louis, MO) per 30 g of body weight, prepared at a concentration of 10 or 20 mg/mL in corn oil (Sigma).

## Slice preparation for ex vivo electrophysiology

$Neurog2^{CreER}$-Tdt mice of either sex aged between post-natal week 4 (W4) and W11 treated with tamoxifen at E12.5, E14.5, or E16.5 were used for experiments. First, they underwent deep anesthesia by intraperitoneal xylazine/ketamine injection (Imalgene 100 mg/kg, Rompun 10 mg/kg) prior to decapitation. A total of 300 μm-thick horizontal slices were cut with a Leica VT1200 Vibratome using the Vibrocheck module in ice-cold oxygenated modified artificial cerebrospinal fluid with the following composition (in mM): 126 CholineCl, 26 $NaHCO_3$, 7 $MgSO_4$, 5 $CaCl_2$, 5 D-glucose, 2.5 KCl, 1.25 $NaH_2PO_4$. Slices were then transferred for rest at room temperature (at least 1 hr) in oxygenated normal aCSF containing (in mM): 126 NaCl, 3.5 KCl, 1.2 $NaH_2PO_4$, 26 $NaHCO_3$, 1.3 $MgCl_2$, 2.0 $CaCl_2$, and 10 D-glucose, for a total of 300 ± 10 mOsm (pH 7.4). A total of 72 cells were successfully recorded from 34 mice (E12.5: n = 22, E14.5: n = 27, E16.5: n = 23). Sixty-three cells were used for morphological analysis.

## Ex vivo patch clamp recordings

Patch clamp recordings in adult slices were performed using a SliceScope Pro 1,000 rig (Scientifica) equipped with a CCD camera (Hamamatsu Orca-05G) and with a X-Cite 120Q (Excelitas Technologies Corp.) fluorescence lamp. Recordings were performed from visually identified Tdt+ cells.

Patch electrodes (4–7 MΩ resistance) were pulled using a PC-10 puller (Narishige) from borosilicate glass capillaries (GC150F10, Harvard Apparatus). Slices were transferred to a submerged recording chamber and continuously perfused with oxygenated ACSF (3 mL/min) at 32 °C. Electrophysiological signals were amplified, low-pass filtered at 2.9 kHz and digitized at 10 kHz with an EPC10 amplifier (HEKA Electronik) and acquired using the dedicated software PatchMaster. Neurons were kept at –65 mV throughout the session and recordings started 5–10 min after access. For current clamp experiments, the following intracellular solution was used (in mM): 130 K-MeSO$_4$, 5 KCl, 5 NaCl, 10 HEPES, 2.5 Mg-ATP, 0.3 GTP, and 0.5 % neurobiotin (~280 mOsm, 7.3 pH). Capacitance compensation and bridge balance were performed before recording and adjusted periodically. Liquid junction potential was not corrected. The resting membrane potential was not measured. For voltage clamp experiments, the solution consisted of (in mM): 130 CsGluconate, 10 HEPES, 5 CsCl, 5 Na$_2$Phosphocreatine, 2 MgATP, 1 EGTA, 0.3 Na$_3$GTP, 0.1 CaCl$_2$ and 5 % neurobiotin ~290 mOsm, 7.3 pH. Uncompensated series resistance was calculated post-acquisition and considered acceptable if below 30 MΩ and if variations were lower than 20 %. Liquid junction potential correction (–13 mV) was applied. Excitatory and inhibitory postsynaptic currents were recorded at –86 mV (E$_{GABA}$) and 0 mV (E$_{Glu}$) respectively, over 9 sweeps of 20 s. A –5 mV step (5 ms duration) was used to monitor series resistance at the beginning of each sweep. For Schaffer collateral stimulation, a bipolar electrode made by two twisted nichrome wires and connected to a DS2A Isolated Voltage Stimulator (Digitimer) was used to deliver 0.2 ms-long stimuli in the *stratum radiatum* of CA1 (between the recorded cell and CA3). Stimulation intensity was set to 2× the minimum intensity capable of eliciting a postsynaptic response (tested between –86 mV and –78 mV). Paired pulse ratio (PPR) was assessed by applying paired 0.2 ms-long pulses (2× minimum stimulation) separated by 50 ms.

## Analysis of ex vivo patch clamp recordings

We have chosen to treat neurons recorded in all patch-clamp experiments as independent replicates. Indeed, only in a few cases were neurons sampled from the same animal with the size of such clustered data varying from one to 4. Taking into account the nested nature of the data would have required sampling from an even number of clusters and objects per cluster in all groups or conditions (*Aarts et al., 2014*). This would constrain too much the experimental design given the low yield of transgenic animals and recombinant cells (Tdt$^+$) and the necessity to minimize the number of mice used. In addition, recordings were performed ex vivo, in different slices for all but three experiments and targeted a precise subpopulation of cells. Analysis of intrinsic membrane properties was performed using custom-made MATLAB scripts (available on https://gitlab.com/cossartlab/cavalieri-et-al-2021.git; *Cavalieri and Cossart, 2021*). Traces were filtered using the sgolay MATLAB function, unless firing properties were being analysed. The input membrane (Rm) resistance was calculated from the slope of steady-state voltage responses to a series of subthreshold current injections lasting 500 ms (from –100 pA to last sweep with a subthreshold response, 10 or 20 pA step size). The membrane time constant ($\tau$m) was estimated from a bi-exponential fit of the voltage response to a small (–20 pA) hyperpolarizing pulse. Capacitance (Cm) was calculated as $\tau$m/Rm. The sag potential was calculated by injecting a 1 s-long negative current step (–200 pA) as follows: $Sag_{-200pA} = \Delta V_{steady-state} - \Delta V_{peak}$ (subtraction of the steady-state potential at halfresponse (averaged over 45 ms) to the minimum peak potential at the beginning of the response). The rebound potential was computed, in absence of rebound action potentials, by subtracting the depolarization peak after the end of the hyperpolarization to the baseline potential: $Rebound_{-200pA} = \Delta V_{peak} - \Delta V_{baseline}$. Firing curves were determined by applying an increasing range of 1 s long depolarizing current injections (up to 500 pA, + 40 pA step). The rheobase (in pA) was defined as the minimum current value necessary to elicit an action potential. The first spike in response to a positive current injection was used to determine: the threshold potential (the peak of the second derivative), the action potential amplitude and fast afterhyperpolarization (both calculated from threshold potential), and the half-width (width at half-amplitude between the threshold potential and the peak of the action potential). Analysis of spontaneous postsynaptic currents (sPSCs) was performed using MiniAnalysis (Synaptosoft). Traces were filtered with a LoPass Butterworth (cutoff frequency 1000 Hz) and PSC amplitude and frequency were calculated over 2 minutes (total of 6 sweeps, 20 s). The mean PSC event was analysed with a custom-made MATLAB script (available on https://gitlab.com/cossartlab/cavalieri-et-al-2021.git) to compute rise time (10%–90% of the ascending phase); decay time (90–37% of the descending phase); decay time constant

$\tau$ (from a monoexponential curve adjusted to best fit the trace); half-width (width at half-amplitude between PSC peak and baseline); area (using trapz MATLAB function). Mean stimulation-evoked PSC amplitude, area and kinetics were calculated as detailed above. In case of multi-peak responses, the decay time constant was measured on the descending phase of the last synaptic event. PPR was measured by calculating the amplitude of two 0.2 ms-long stimuli at 50 Hz as follow: PPR = peak2/peak1. The amplitude of each response was measured by averaging up to 10 sweeps and measuring the maximum peak from baseline, calculated between the stimulation time and the start of the current response ascending phase. The ratio between excitatory and inhibitory PSC (E/I ratio) was calculated by dividing frequency and amplitude values.

## Morphological analysis of neurobiotin-filled cells

Slices containing neurobiotin-filled cells were fixed overnight at 4 °C in PFA (4%), rinsed in PBS containing 0.3 % Triton X-100 (PBS-T) and incubated overnight at room temperature in streptavidin-AlexaFluor488 (1:1,000 in PBS-T). Slices were mounted using Vectashield mount medium containing DAPI (VectorLabs). Post-hoc analysis was performed using an AxioImager Z2 microscope equipped with Apotome module (Zeiss). The co-localization of neurobiotin and TdTomato was verified systematically for every recorded cell. Neurobiotin-filled neurons were selected when the apical dendrites were clearly visible and uncut. Stacks of optical sections were collected for these cells. The primary dendrite was identified as the portion of between the soma and a bifurcation generating secondary dendrites of roughly equal size. The length of the primary dendrite was measured by approximating its shape to 1–3 linear segments.

## Quantification of cell location and abundance in CA1 *stratum pyramidale*

For patch clamp recordings, the cell location was measured from the border between the *stratum pyramidale* (SP) and the *stratum radiatum* (SR) and normalized to the thickness of the SP, meaning that values close to 0 correspond to the proximity of SP/SR border and values close to one to *stratum oriens* (SO)/SP border. For the overall quantification of *Neurog2* cell location in *Figure 1*, *Figure 4—source data 1Figure 5—figure supplement 2* and *Figure 7—source data 1*, *Figure 8—figure supplement 1*, the cell location was measured from the border between the SP and the SR and expressed in micrometers. To calculate the relative abundance of the three fate-mapped cell groups in CA1, we used a population approach to quantify Tdt+ neurons in the whole brain of tamoxifen-induced *Neurog2CreER*-Tdt adult mice. In brief, whole brains were clarified using the CUBIC method (*Susaki et al., 2014*) and imaged with light-sheet microscopy (ultramicroscope II, Lavision Biotec). Then, 3D reconstructions were computed and Tdt+ PN distribution was registered creating a single-cell level cartography of fate-mapped glutamatergic neurons in the whole brain. Finally, using a custom-made MATLAB script Tdt+ cell maps were compared to the Allen Brain Atlas annotations, those included in the whole CA1 area were counted and the biggest population (E14.5 PNs) was used to calculate the relative abundance.

## Histological processing and immunohistochemistry

After deep anesthesia with a ketamine (250 mg/kg) and xylazine (25 mg/kg) solution (i.p.), animals were transcardially perfused (1 mL/g) with 4 % paraformaldehyde in saline phosphate buffer (PBS). Brains were post-fixed overnight, then washed in PBS. For Ctb tracing, a VT1200 Vibratome (Leica) was used to cut coronal slices. Slice thickness was 100 µm for the injection site and 70 µm for the hippocampus. In a subset of experiments, Ctb-AlexaFluor647 signal was amplified by immunohistochemistry. Sections were incubated overnight with primary Cholera Toxin beta polyclonal antibody (rabbit; 1:1000, RRID:AB_779810, Invitrogen) diluted in PBS containing 0.3 % Triton X (PBS-T) and then for approximately 1h30 with donkey anti-rabbit secondary antibody AlexaFluor647 (1:500, Jackson Immunoresearch, RRID:AB_2492288) in PBS-T. For cFos immunostaining, a similar procedure was employed. Coronal slices (50 µm thickness) obtained with a HM450 sliding Microtome (Thermo Scientific, Waltham, MA) were rinsed three times in PBS-T and incubated overnight at 4 ° C with a solution containing 5 % goat serum (GS) and anti-cFos (rabbit; 1:5000, RRID:AB_190289, Abcam) diluted in PBS-T. The following day slices were exposed to a goat anti-rabbit AlexaFluor647 secondary antibody (1:500, Jackson Immunoresearch, RRID:AB_2338072) in PBS-T. Post-hoc analysis was performed

from image stacks (1.5 µm interval, 7 images) obtained using a Zeiss LSM-800 system. Slices were mounted using Vectashield mount medium containing DAPI (VectorLabs). For calbindin (CB) staining, 70 µm slices were incubated overnight at 4 ° C with 5 % normal donkey serum (NDS) and anti-CBD28k primary antibody (rabbit; 1:1000, RRID:AB_10000340, Swant) diluted in PBS-T, followed by incubation with donkey anti-rabbit secondary antibody AlexaFluor647 (1:500, 2 hr). Post-hoc analysis was performed from image stacks obtained using a Zeiss LSM-800 system equipped with emission spectral detection and a tunable laser providing excitation range from 470 to 670 nm.

## Quantification of putative PV-expressing synaptic boutons

For parvalbumin (PV) putative puncta detection, we employed a similar immunostaining procedure as described above. *Neurog2^{CreER}*-Tdt adult mice (age> P50) of either sex were used. Goat anti-PV primary antibody (1:1000, Swant, pvg-214, RRID:AB_10000345) and donkey anti-goat AlexaFluor488 secondary antibody (1:500, Jackson Immunoresearch, RRID:AB_2336933) were used. To compare the PV perisomatic innervation across CA1PNs with different birthdates, image stacks (0.06 × 0.06 × 0.410 µm) centered on the soma of TdTomato + cells (E12.5 CA1PNs: 88 cells; E14.5 CA1PNs: 106 cells; E16.5 CA1PNs: 44 cells) were acquired with a Zeiss LSM-800 microscope using a Plan-Apo 40 x/1.4 oil objective. The mean total thickness of the stacks was 15,85 µm, consistent with the mean soma diameter of pyramidal cells in adult CA1 (*Ishizuka et al., 1990*). Volume overlaps between PV+ boutons and Tdt⁺ somata were calculated using a custom-made MATLAB script. Values were normalized by the number of Z-steps to control for possible differences in soma size or experimental variability.

## Stereotaxic injections for retrograde tracing

*Neurog2^{CreER}*-Tdt adult mice (age> P50) of either sex were anaesthetized using 1–3% isoflurane in oxygen. Analgesia was provided with buprenorphine (Buprecare, 0.1 mg/kg). Lidocaine was applied by cream or subcutaneous injection before the incision for additional local analgesia. Mice were fixed to a stereotaxic frame with a digital display console (Kopf, Model 940). Under aseptic conditions, an incision was made in the scalp, the skull was exposed, and a small craniotomy was drilled over the target brain region. A 200–400 nl volume of 0.1 % AlexaFluor647-conjugate Cholera Toxin subunit b (Ctb, Thermofisher Scientific) was delivered using a glass pipette pulled from borosilicate glass (3.5″ 3-000-203-G/X, Drummond Scientific) and connected to a Nanoject III system (Drummond Scientific). The tip of the pipette was broken to achieve an opening with an internal diameter of 30–40 µm. Stereotaxic coordinates were based on a mouse brain atlas (Paxinos and Franklin, 3rd edition). All coordinates are indicated in *Table 1* in millimeters, and examples of the histological recovery of the injection sites are displayed in *Figure 5—figure supplement 1*. Antero-posterior (AP) coordinates are relative to bregma; medio-lateral (ML) coordinates are relative to the sagittal suture; dorso-ventral (DV) coordinates are measured from the brain surface. Mice were perfused 12–15 days later, to allow sufficient Ctb uptake and transport from the synaptic terminals.

## Analysis of retrograde tracing

**Table 1.** Stereotaxic coordinates of CA1PNs target regions.

| Target region | AP | ML | DV |
|---|---|---|---|
| NAcc | + 1.8/ + 2.0 | −0.6/−0.45 | −4.0 |
| Amy | −1.4 | −3.3 | −4.3 |
| mPFC | + 2.0 | −0.35 | −1.8 |
| LHA | 1.34/−1.45 | −0.65/−1.0 | −4.8/−4.0 |
| LS | + 0.6/0.7 | −0.4/−0.35 | −2.6/2.55 |

NAcc, Nucleus Accumbens (shell), Amy: Amygdala; mPFC, medial prefrontal cortex; LHA, lateral hypothalamic area, LS: lateral septum.

To confirm that injections were successful, each injection site was visually inspected and compared to the Allen mouse brain atlas. The occurrence of co-labeling with Ctb among Tdt⁺ cells was verified and the ratio of Ctb⁺/TdT⁺ cells calculated manually. This resulted in the generation for each injected animal of a binary vector of length L equal to the total count of Tdt⁺ cells and as many '1' entries as the number of identified Ctb⁺/TdT⁺ cells. Then, vectors corresponding to data from animals labeling CA1PNs with the same birthdate and injected with Ctb in the same target region were pooled together in a 'group vector'. Finally, a bootstrap resampling approach was used to compute the pairwise comparisons (total number

**Table 2.** Summary of adult *Neurog2*[CreER]-Tdt mice used in post-exploration cFos analysis.

| N° animals | HC | FAM | NOV | Total |
|---|---|---|---|---|
| E12.5 | 2 | 2 | 3 | 7 |
| E14.5 | 3 | 3 | 4 | 10 |
| E16.5 | 2 | 3 | 3 | 8 |
| Total | 7 | 8 | 10 | 25 |

of tests = 45). In brief, each of two group vectors A and B were randomly resampled with replacement for 10,000 times. At each iteration, the difference between the two group vector means ($\Delta_\mu = \mu_A - \mu_B$) was calculated and stored. At the end of the resampling, the 99.9 % confidence interval (CI) of the distribution of ($\Delta_\mu$) was computed. The difference between group vectors A and B was considered significant if the CI did not include the value 0.

## cFos expression upon exploration

A total of 25 tamoxifen-treated *Neurog2*[CreER]-Tdt mice between 10 and 12 weeks old of either sex were used. Prior to the experiment, animals were single-housed and divided in three groups, named home-cage (HC), familiar (FAM) and novel (NOV), see *Table 2*. Group HC was carried to the experimental room, handled by the experimenter during 5–10 min for 3 consecutive days (D1, D2, D3) and perfused 1 hour after handling on D3. Group FAM was exposed to an exploration chamber for 20 min from D1 to D3, and returned to their home cage immediately after. Group NOV was handled during D1 and D2 and left in the exploration chamber for one single 20 -min session on D3. FAM and NOV were both perfused 1 hour after exploration. The chamber consisted of a transparent plastic rat cage sized 435 × 290 mm containing visual, tactile, and olfactory (butanal) cues. The center zone was defined as a 290 × 193 mm rectangular area at the center of the box. A white noise (20/30 dB) was played in the experimental room for the duration of the exploration and low lighting (~25 lux) was centered over the box. Mice position during each session was recorded with a Basler Ace camera (Basler AG, Ahrensburg) and tracked with EthoVision XT 11 (Noldus, Leesburg, VA) software.

## Analysis of cFos expression upon exploration

Tdtomato[+] somata were segmented with Fiji Trainable WEKA Segmentation plugin. Two custom-made Cellprofiler pipelines were used on DAPI and AlexaFluor647 (cFos) channels (*Carpenter et al., 2006*). (i) To provide an estimation of DAPI cell density, a single 2D image (from the middle of the stack) was segmented using a Minimum Cross Entropy threshold. (ii) Segmentation of cFos[+] nuclei was first achieved by applying the Robust Background thresholding method on each single 2D image composing the stack and converting them into binary masks. Then, these masks were restacked and somas were identified with the 3D Object Counter Fiji plugin across the depth of the stack. To determine TdTomato and cFos colocalization, a custom-made MATLAB script was used. In a nutshell, matrices representing TdTomato and cFos binary masks were multiplied, generating a new binary image of areas presenting overlap between the two channels. Finally, each of these areas, which represented putative colocalized cells, was manually inspected for confirmation. At the end of this procedure, we applied the same procedure as described for the analysis of retrograde tracing. We computed for each animal a binary vector of length L equal to the total number of segmented Tdt[+] cells and as many '1' entries as the number of identified cFos[+]/Tdt[+] cells. Then, vectors corresponding to mice in the same birthdate group and same behavioral condition were pooled in a 'group vector'. Finally, a bootstrap resampling approach was used to compute the pairwise comparisons (total number of tests = 18). The 95 % confidence interval (CI) of the distribution of the bootstrapped differences of means ($\Delta_\mu$) was computed. The difference between group vectors A and B was considered significant if the CI did not include the value 0.

## Statistical analyses

Statistical analysis was done using Prism (GraphPad) and custom-made MATLAB scripts. In patch clamp experiments, outliers were removed (ROUT method, Q = 0.1%). For comparing input-output firing curves, two-way ANOVA was used. When comparing birthdate groups for a given measure, median-based bootstrap resampling was used to compute pairwise comparisons, subsequently corrected with Holm-Bonferroni method to control for family-wise error rate, except for morphological measures where mean and Sidak's correction were used instead. The correlation with cell location was tested using bootstrap resampling. In the analysis of exploratory

behavior, repeated-measures and ordinary one-way ANOVA were used to compare distance run in FAM condition and FAM vs NOV, respectively. The volume overlap of putative PV boutons and percentage of $CB^+$-PNs are tested among birthdates with one-way ANOVA with Tukey's correction for multiple comparisons. A table including all statistical tests performed is included in the supplementary material as .

## Acknowledgements

This work was supported by the European Research Council under the European Union's Horizon 2020 research and innovation programs (grant agreement no. 646925) as well as by the Agence Nationale pour la Recherche (ANR, Programme Blanc bilatéraux, ANR-13-ISV40002-01 "EbGluNet" and HIPPOPLAST, JTC-2017–021) and by the Fondation Bettencourt Schueller (Prix des Sciences de la Vie). AB and RC are supported by the Centre National de la Recherche Scientifique (CNRS). We thank Pr. David Anderson for providing the *Neurog2*$^{CreER}$ mouse line. We thank L Cagnacci for technical support and S Pellegrino-Corby, M Kurz and F Michel from the INMED animal and imaging facilities (InMagic). We are grateful to Dr. S Reichinnek for valuable scientific input, to Dr. S Sarno for her help on statistical analysis.

## Additional information

### Funding

| Funder | Grant reference number | Author |
|---|---|---|
| H2020 European Research Council | 646925 | Rosa Cossart |
| Agence Nationale de la Recherche | ANR-13-ISV40002-01 | Rosa Cossart |
| Agence Nationale de la Recherche | JTC-2017-021 | Rosa Cossart |
| Fondation Bettencourt Schueller | Prix des Sciences de la Vie | Rosa Cossart |

The funders had no role in study design, data collection and interpretation, or the decision to submit the work for publication.

### Author contributions

Davide Cavalieri, Conceptualization, Formal analysis, Investigation, Methodology, Writing - original draft; Alexandra Angelova, Anas Islah, Investigation, Methodology; Catherine Lopez, Agnès Baude, Investigation; Marco Bocchio, Methodology, Supervision; Yannick Bollmann, Formal analysis, Software; Rosa Cossart, Conceptualization, Funding acquisition, Methodology, Project administration, Resources, Supervision, Writing - original draft, Writing - review and editing

### Author ORCIDs

Davide Cavalieri (iD) http://orcid.org/0000-0003-2369-647X
Marco Bocchio (iD) http://orcid.org/0000-0002-7005-3453
Agnès Baude (iD) http://orcid.org/0000-0002-7025-364X
Rosa Cossart (iD) http://orcid.org/0000-0003-2111-6638

### Ethics

All protocols were performed under the guidelines of the French National Ethics Committee for Sciences and Health report on "Ethical Principles for Animal Experimentation" in agreement with the European Community Directive 86/609/EEC under agreement #01 413.03. All efforts were made to minimize pain and suffering and to reduce the number of animals used.

### Decision letter and Author response

Decision letter https://doi.org/10.7554/eLife.69270.sa1
Author response https://doi.org/10.7554/eLife.69270.sa2

# Additional files

## Supplementary files
- Transparent reporting form
- Supplementary file 1. Statistics table.

## Data availability
Data generated or analysed during this study are included in the manuscript or available on Dryad (https://doi.org/10.5061/dryad.76hdr7swh). A single source data file (multiple sheets) is included in the submission. Raw data and code from the following experiments are also included in the submission: ex vivo electrophysiology.

The following dataset was generated:

| Author(s) | Year | Dataset title | Dataset URL | Database and Identifier |
|---|---|---|---|---|
| Cavalieri D | 2021 | Whole-cell current clamp recordings in slice | http://dx.doi.org/10.5061/dryad.76hdr7swh | Dryad Digital Repository, 10.5061/dryad.76hdr7swh |
| Davide C, Alexandra A | 2021 | Anti-cfos immunostaining after exploration | https://doi.org/10.5061/dryad.280gb5mq2 | Dryad Digital Repository, 10.5061/dryad.280gb5mq2 |
| Davide C, Alexandra A | 2021 | Exploration data for cfos expression analysis | https://doi.org/10.5061/dryad.7d7wm37v2 | Dryad Digital Repository, 10.5061/dryad.7d7wm37v2 |
| Cavalieri D | 2021 | Whole-cell voltage clamp recordings in slice | https://doi.org/10.5061/dryad.r4xgxd2cf | Dryad Digital Repository, 10.5061/dryad.r4xgxd2cf |

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
