## [Editor Report]

The authors use powerful fate-mapping and complementary neuroanatomical and electrophysiological approaches to examine development of hippocampal circuits. They identify a new group of pioneer CA1 pyramidal neurons, which are born early in embryonic development, contribute to CA1 neuronal diversity, and play a role in encoding familiar environments.

---

## [Decision Letter]

**Decision letter after peer review:**

Thank you for submitting your article "CA1 pyramidal cell diversity is rooted in the time of neurogenesis" for consideration by *eLife*. Your article has been reviewed by 3 peer reviewers, including Marco Capogna as Reviewing Editor and Reviewer #1, and the evaluation has been overseen by John Huguenard as the Senior Editor.

Essential revisions:

All reviewers agree that the major strength of the manuscript is represented by the information that birthdate is a strong predictor of adult CA1 PNs diversity All the reviewers, however, have also raised criticism that needs to be addressed. One weakness of the manuscript is that it mostly remains at descriptive and correlative levels. Furthermore, the claim that the early, pioneer CA1 pyramidal cells show unique feature should be modified based on the experimental evidence that rather show a continuum of properties of cells born at subsequent embryonic days. Finally, there are a number of issues related to statistics, rigor of analysis, methodological details, and sample sizes that need to improve.

The most important points that the reviewers suggest to address can be summarized as following.

1) Mechanism. The manuscript lacks any insight on the potential mechanisms underlying the influence of birthdate on subsequent CA1 PNs diversity. Reviewer 1 acknowledges that this is an ambitious task that could be hard to include this important point in this manuscript. However, Reviewer 1 would like to see some attempts toward this direction already in the next edition of the manuscript, because if positive, this would greatly increase the impact of the manuscript. Some ideas toward this goal are given by reviewer 1(#1).

2) Behavior. At present the Authors probed ventral CA1 PNs by using a familiarity-novelty behavioral test. However, it would a significant improvement to corroborate this point and use an anxiety-like behavioral test that better fit the ventral hippocampus' connectivity and function. This aspect should be addressed by performing new experiments or at least by better discussing the possibility that ventral CA1 early born PNs may have a role on anxiety based on more stringent test as for example illustrated by Fredes et al., Curr Biol 2021, Kheirbel…Hen, Neuron (2013). It would be also desirable to corroborate the claim of the putative role of early born PNs "in the consolidation or retrieval of recent experience", because the data show effects mainly on learning and not memory retrieval.

3) Statistic and sample size. Increased rigor should be integrated into the statistics presented, as explained by reviewer 2 (#1, 2, 3 and 5) and also in many sections of reviewer's 3 report.

4) Electrophysiological data. Key improvements are proposed by reviewers 2 and 3. These regard for example, analysis of the paired-pulse synaptic data, definition of synaptic integration properties of the three different cohorts of fate-mapped neurons, analysis of spontaneous synaptic events.

5) Fate mapping strategy. In the revised version, the Authors should go beyond the limits of the fate-mapping approach by using BrdU labeling for their tracing and behavioral/c-fos data, as explained by reviewer 3.

6) Key editorial changes. Several important editorial changes are suggested by all the reviewers. For example, the revised manuscript would significantly improve when the claim that pioneer CA1 PNs display "unique features" in the results and Discussion sections would be changed to better reflect the data that mostly show a continuum of properties between early and late born cells.

7) Injection sites. The specificity for the injection sites should be presented in a more complete way by adding more examples.

*Reviewer #1 (Recommendations for the authors):*

In my opinion the quality of the data and text submitted (text, figures, data analysis, methods, and discussion) is promising. However, the manuscript should be revised and made it less descriptive for example by attempting to provide some evidence of the underlying mechanisms and by testing or at least discussing behavioral roles, e.g. anxiety, more specific for the ventral hippocampus.

Main comments

1) The Authors should investigate the mechanisms underlying the influence of birthdate on subsequent CA1 PNs diversity. For example, the data submitted already allow to advance the idea that the main source of activation for both E12.5 nd E16.5 CA1 PNs works through synaptic glutamatergic excitation. However, why not to test this idea directly? For example, what happens to the fate mapping of these neuronal populations in the presence of excitatory amino acid antagonists or blocking excitatory synaptic activity? A complementary idea advanced by the Authors is that E12.5CA1PNs receive weak somatic inhibition. Again, this hypothesis could be tested by observing the effects of strengthening synaptic inhibition on the fate mapping of this cell group.

2) At present the Authors investigated the activation of birth dated ventral CA1 PNs by using a familiarity-novelty behavioral test. I am not convinced that this test represents an optimal choice. One possibility is that the Authors would re-work on this protocol and use an anxiety-like behavioral test that according to several published papers (e.g., Fredes et al., Curr Biol 2021, Kheirbel…Hen, Neuron (2013)) is more directly related to the ventral hippocampus. It would be also desirable to corroborate the claim of the putative role of early born PNs "in the consolidation or retrieval of recent experience", because the data show effects mainly on learning and not memory retrieval.

*Reviewer #2 (Recommendations for the authors):*

1. Increased rigor should be integrated into the statistics presented. For most measures, the authors describe how a "median-based bootstrap resampling was used to compute pairwise comparisons, subsequently corrected with Holm-Bonferroni method to control for family-wise error rate". While bootstrapping approaches are useful for characterizing distributions, it is not clear how the authors tested the null hypothesis that all groups had identical median values. The authors also need to appropriately account for datasets where nested data includes multiple measurements made within the same biological replicate (e.g. multiple cells patched from the same animal). Aarts et al., 2014 (PMID: 24671065) provides a good overview of this issue. Throughout the manuscript, information such as F-statistics, degrees of freedom, number of samples per group (rather than just overall for the experiment), and p-values for non-significant comparisons should be added. Lastly, the authors should exercise caution in attributing conclusions to non-significant p-values (i.e. Figure 4B and Figure 7C2).

2. The authors should include some quantification of the relative abundance of the different cell populations in Figure 1. As E12.5 CA1PNs appear considerably rarer than E14.5 or E16.5 neurons, yet are sampled at equal levels in most of the other assays, the authors might want to consider weighting their regression analyses to account for this disparity. Contextualizing the Tdtomato labeling in Figure 1 with a more classical marker of deep vs superficial CA1PN populations, such as Calbindin, would strengthen the findings.

3. The distribution of E12.5 CA1PNs sampled in Figures 2-3 appears to differ substantially from those in Figures 1 and 5 6. While Figure 1 shows that almost no E12.5 neurons are located in SR and many are found in SO, the E12.5 populations recorded from in Figures 2-3 include multiple cells from SR and none from SO. The authors should address how this non-representative sampling could alter their findings related to the synaptic signaling properties of the different CA1PN populations.

4. Details of how paired pulse facilitation is calculated in Figure 3 should be further clarified. Since there are multiple peaks in the elicited responses, the authors should make sure to only count the first peak that appears. In addition, due to the differences in decay and that the second pulse comes before the first pulse response returns to baseline, the amplitude of the second peak needs to account for the differences in current amplitude when the second pulse occurs. This issue could be mitigated if the authors had used a stimulation interval of 40ms instead. The authors should also consider how accurately they can assess decay times given the prevalence of multi-peak responses.

5. The number of samples is very low in Figure 4, with most regions in E12.5 animals having an n=2. This likely contributes substantially to the low power to differentiate between regions in Figure 4C. In light of this, it is difficult to give much weight to the conclusion that E12.5 CA1PNs project equivalently to all of the assessed regions. Adding a similar analysis taking into account the position of Ctb+/Tdt+ CA1PNs within the pyramidal layer to Figure 4, as is done in most of the other figures, would also be insightful. The cFos measurements in Figure 7 appear to be similarly underpowered.

6. While the description of the E12.5 CA1PN pioneer population is intriguing, it is unclear whether this relatively sparse population not following the same trends as between E14.5 and E16.5 neurons convincingly disproves the importance of birth order in predicting CA1PN diversity. For many traits, E12.5 and E16.5 neurons show considerable similarity, suggesting that they may just be early and late emerging populations of deep CA1PNs.

*Reviewer #3 (Recommendations for the authors):*

This is an interesting study by Cavalieri et al., examining the birth date-specific electrophysiological features, wiring and function of subpopulations of CA1 pyramidal cells of the rodent hippocampus. It presents a set of stimulating facts about neurons fate mapped at E12.5, E14.5 and E16.5 that can have important implications on the role of development on adult function of circuits. I have a number of suggestions that I believe would strengthen the findings of the study that the authors may want to consider.

1) The fate-mapping strategy that has been successfully employed by the researchers in this and previous studies allows them to mark cells that are becoming post-mitotic at the three developmental time points that they have selected. Nevertheless because of the limitations of the method, only a handful of cells born at a given time point are labeled and this is especially true for the E12.5 fate-mapped neurons. This precludes the collection of extensive datasets and the opportunity to make robust statements on the stratification of CA1 pyramidal cells based on birth date versus location. One way this limitation could be overcome for two of the main and most novel findings the researchers have obtained, namely the tracing and behavioral/c-fos data, is by utilizing BrdU labeling. The researchers could perform BrdU labeling at the three different embryonic time points, followed by the tracing and c-fos/behavioral experiments to show that the more extensive data they will get matches the ones obtained with Ngn2CreER.

2) The representative examples of the recording sEPSC shown in figure 2 are very noisy. It would therefore be of added value if the authors showed examples of which of the peaks are detected as events by the criteria and pipeline they are using.

In addition it would be good if they also showed representative examples of the sIPSCs in the figure and reported how they detected peaks for those too.

3) On the bottom of page 12 the authors make the following statement: "Interestingly, none of the intrinsic electrophysiological properties measured here correlated with the location in the stratum pyramidale, suggesting that the embryonic birth date is a major determinant of the observed cell heterogeneity, which cannot solely be explained by the radial gradient".

The reviewer finds this to be a strong statement based on the data the authors show. There are two points from the E16.5 group that are closer to the SO and they have overlapping values with the E12.5 group. In general, since the authors show that birth date correlates well with positioning, making a claim that the position does not matter, but it is the birth date that does, reads somewhat contradictory.

Along the same lines and as it regards figure 2C the authors claim that: in contrast, the frequency of s-EPSCs (Figure 2C) showed a linear correlation with the cell body location (r = 0.49, p: 0.002). This was also reflected in the E/I frequency ratio (r=0.43, P: 0.0047), indicating a higher excitatory synaptic drive in deep than in superficial neurons, regardless of their birthdate (Figure 2E).

This statement would suggest that it is the position that determines the features of the inputs and not the birth date.

In general, it would be informative, since this one of the main points of the study, if the authors showed which point reported in the graphs corresponds to which cell as it regards its position in CA1. Adding more cells from the tails of the spatial distribution of the cohorts would strengthen the claim that the important factors here is birth date and not position.

4) In terms of the data presented in figure 3. The manuscript would be further improved if the authors looked at the integration capacities of the different 3 cohorts of fate-mapped cells using current clamp and SC stimulation. Also, since it is shown that there is a difference in the perisomatic PV-positive inhibitory synapses the E14.5 ones receive, why not perform stimulation of the pyramidal cells layer to record evoked IPSCs coming from these synapses instead of the SC?

5) In many of the figures (including figure 3) the all the individual data points are presented, but with no mention on the actual number of points and that of biological replicates. It would be good for the reader to have these numbers indicated in the panels of the figure or in the figure legend, since it is important for the statistics applied.

6) Along the same lines, in table 2 the authors report the number of mice used for each of the behavioral experiments with a number of conditions only having "2" animals. I think the results would be more robust if the authors increased the number for each of the conditions to at least "3".

7) Regarding the data presented in figure 5 D1 and D2 on the sag at -200pA. The data here is clear, but it would also be informative to report whether the sag is different if the different groups are hyperpolarized at the same potential rather than using the same magnitude current pulse. It is expected that cells that have a higher input resistance (E14.5) would show a higher hyperpolarization upon the same current pulse. I would suggest that the authors inject enough current to hyperpolarize the cells for each group to the same value and assess if the sag is indeed different between the groups.

8) In figure 7B2 one would potentially expect that there would be a significant increase in the labeling of cells in NOV over FAM environment, since it has been previously shown that the amount of activation of cells in a more familiar environment decreases compared to a novel one. This would also correlate well with the decreased exploratory behavior of the mice in this experiment. The authors should make a comment on this point and on how their c-fos expression may correlate with prior spiking activity and over which time scales.

9) Since one of the most interesting findings of the study is the differential proportion of output nuclei shown for some of the cohorts, it is important that the specificity of the injection sites is presented in the most convincing manner. In this respect the panel in Supplementary figure 1 C targeting the lateral septum is not as clear, since there seem to be many fibers labeled, which also makes it hard to spot where the injection site is. Perhaps adding the respective digital atlas next to each image and a close up insert of the injection site would be helpful for the readers to navigate the position of the injections.

---

## [Author Response]

Reviewer #1 (Recommendations for the authors):In my opinion the quality of the data and text submitted (text, figures, data analysis, methods, and discussion) is promising. However, the manuscript should be revised and made it less descriptive for example by attempting to provide some evidence of the underlying mechanisms and by testing or at least discussing behavioral roles, e.g. anxiety, more specific for the ventral hippocampus.Main comments1) The Authors should investigate the mechanisms underlying the influence of birthdate on subsequent CA1 PNs diversity. For example, the data submitted already allow to advance the idea that the main source of activation for both E12.5 nd E16.5 CA1 PNs works through synaptic glutamatergic excitation. However, why not to test this idea directly? For example, what happens to the fate mapping of these neuronal populations in the presence of excitatory amino acid antagonists or blocking excitatory synaptic activity? A complementary idea advanced by the Authors is that E12.5CA1PNs receive weak somatic inhibition. Again, this hypothesis could be tested by observing the effects of strengthening synaptic inhibition on the fate mapping of this cell group.

We thank the reviewer for bringing up this important issue. Addressing the mechanisms by which birthdate influences diversity is a huge task, not even yet solved for any subtype of GABAergic neuron, which diversity, also rooted at their embryonic origin, has been investigated for decades by many groups. In our specific case, this question would require being able to manipulate birthdated neurons. Unfortunately, our cells are barely accessible to manipulation, since they only express Cre transiently in the embryo. We have been trying to find ways to achieve this goal for many years, including through the design of a mouse line that would convert the transient expression of Cre into a permanent expression of flp. However, we are still in the process of testing the specificity of the flp-dependent viruses, which seems not as straightforward as we had anticipated. Besides specific genetic manipulation of birthdated cells, we do not see any experiment that would address this question without perturbing the entire CA1 network, but we will be happy to hear any suggestion. We have now mentioned this technical limitation in the discussion paragraph (pages 19, 20 and 22).

2) At present the Authors investigated the activation of birth dated ventral CA1 PNs by using a familiarity-novelty behavioral test. I am not convinced that this test represents an optimal choice. One possibility is that the Authors would re-work on this protocol and use an anxiety-like behavioral test that according to several published papers (e.g., Fredes et al., Curr Biol 2021, Kheirbel…Hen, Neuron (2013)) is more directly related to the ventral hippocampus. It would be also desirable to corroborate the claim of the putative role of early born PNs "in the consolidation or retrieval of recent experience", because the data show effects mainly on learning and not memory retrieval.

We agree with the reviewer that an anxiety-like protocol would be an interesting additional behavioral test. However, as explained above in our preliminary remark, these experiments would require at least 4 months, if not a year (the time spent for our previous behavioral assays, including the long analysis by segmentation of the labeled slices). That said, in order to address this point, we reasoned that the time spent in the center of the arena in our previous experiments could indirectly provide some measure of the level of anxiety.

We therefore decided to perform additional analysis on the thigmotaxis, as an index of anxiety-like state during exploration. We found no linear correlation between cFos expression rate and the percentage of time spent exploring the central part (more anxiogenic) of the rectangular arena. This suggests that our protocol is not well suited for probing anxiety-like behavior through the analysis of cFos activation.

Reviewer #2 (Recommendations for the authors):1. Increased rigor should be integrated into the statistics presented. For most measures, the authors describe how a "median-based bootstrap resampling was used to compute pairwise comparisons, subsequently corrected with Holm-Bonferroni method to control for family-wise error rate". While bootstrapping approaches are useful for characterizing distributions, it is not clear how the authors tested the null hypothesis that all groups had identical median values.

We thank the reviewer for the helpful input. On this matter, we would like to motivate our choice of not using a classic null-hypothesis significance testing and explain why, to our understanding, it is appropriate.

First, for most variables tested, we could not opt for a parametric approach. Second, non-parametric

Kruskal-Wallis test has little power and loses precision for small sample sizes (more likely to confirm the global null hypothesis H0). Third, resampling approaches (bootstrap) do not require any assumption on the sample distribution and are suitable for small sample sizes. Fourth, for the tracing and cfos experiments, given the limited number of animals, we used samples issued from discrete data (0/1 vectors) that are not compatible with either ANOVA or Kruskal-Wallis test.

We believe that the analytical approach used in the manuscript is coherent with the data used, knowing that we made sure to exclude any possible random inflation of the significance by correcting for multiple comparisons. In addition, we tested (see below) the morphometric and electrophysiological data with ANOVA plus Tukey’s test for stratified data. We found that we could reject H0 in all cases, except membrane tau (absent from the new version of the manuscript), spontaneous synaptic E-I ratio and eIPSC PPR, showing that the two approaches yield very similar results.

The authors also need to appropriately account for datasets where nested data includes multiple measurements made within the same biological replicate (e.g. multiple cells patched from the same animal).

We considered individual neurons as replicates because this is traditionally done in in vitro patch clamp experiments. The rationale of this is that individual patched neurons can be considered somehow independent (although neurons should be sampled from an even and sufficient number of mice across groups). We believe that using neurons as replicates is less problematic in patch clamp experiments than multi-neuron recordings.

Nonetheless, we decided to control for the nested structure of the morphometric and electrophysiological data by performing one-way ANOVAs followed by a Tukey post hoc test with stratified bootstrap (with mouse ID as factor for the stratification). This method yielded very similar results to our previous analyses.

Aarts et al., 2014 (PMID: 24671065) provides a good overview of this issue. Throughout the manuscript, information such as F-statistics, degrees of freedom, number of samples per group (rather than just overall for the experiment), and p-values for non-significant comparisons should be added.

We thank the reviewer for this important remark, this information is now provided throughout the text. Non-significant p-values were also added in the figure legends (except figures 4C, D and 7C2, due to the large number of comparisons).

Lastly, the authors should exercise caution in attributing conclusions to non-significant p-values (i.e. Figure 4B and Figure 7C2).

We thank the reviewer for pointing this out. We agree and have removed any statement regarding non-significant p values both in the main text and legends.

2. The authors should include some quantification of the relative abundance of the different cell populations in Figure 1.

We thank the reviewer for an important point that we have chosen to address using clarified brain samples with CUBIC method (Susaki et al., 2014) and imaged with light-sheet microscopy. With a custom-made MATLAB script, we obtained a quantitative cartography of fate mapped PNs according to the Allen brain atlas registration. We have calculated the numbers of E12.5 (n=4 brains), E14.5 (n=3) and E16.5 (n=3) PNs in the whole CA1 area and found more than seven times fewer E12.5 CA1 PNs per brain (917 neurons ± 851; mean ± SD) than E14.5 neurons (6810 neurons ± 3454), and four times less than E16.5 CA1 PNs (3585 ± 401). These important quantifications are now added to the revised manuscript.

As E12.5 CA1PNs appear considerably rarer than E14.5 or E16.5 neurons, yet are sampled at equal levels in most of the other assays, the authors might want to consider weighting their regression analyses to account for this disparity.

We thank the reviewer for this interesting suggestion. Using the relative cell abundance described in the previous point (E12.5 = 0.135, E14.5 = 1, E16.5 = 0.52), we calculated weighted linear regressions on SPSS software and found very similar results to those present in the manuscript.

Contextualizing the Tdtomato labeling in Figure 1 with a more classical marker of deep vs superficial CA1PN populations, such as Calbindin, would strengthen the findings.

We thank the reviewer for this great suggestion and have performed immunostaining against calbindin (CB) in fixed slices from fate-mapped mice from all birthdate groups. Interestingly, we found that E12CA1PNs were more similar to E16CA1PNs in their CB expression than E14CA1PNs. Indeed, they were more likely to express CB, this even when located in the deep CA1 layer.

3. The distribution of E12.5 CA1PNs sampled in Figures 2-3 appears to differ substantially from those in Figures 1 and 5 6. While Figure 1 shows that almost no E12.5 neurons are located in SR and many are found in SO, the E12.5 populations recorded from in Figures 2-3 include multiple cells from SR and none from SO. The authors should address how this non-representative sampling could alter their findings related to the synaptic signaling properties of the different CA1PN populations.

We thank the reviewer for this detailed analysis of our data and pointing out this difference. To address this criticism, we have analyzed the linear correlations displayed in Figures 2 and 3 removing the E12 CA1PNs located in the SR and found that the conclusions remained unchanged.

4. Details of how paired pulse facilitation is calculated in Figure 3 should be further clarified.

We apologize for this lack of clarity. Please note that a more detailed explanation is now inserted in the method section.

5. The number of samples is very low in Figure 4, with most regions in E12.5 animals having an n=2. This likely contributes substantially to the low power to differentiate between regions in Figure 4C. In light of this, it is difficult to give much weight to the conclusion that E12.5 CA1PNs project equivalently to all of the assessed regions. Adding a similar analysis taking into account the position of Ctb+/Tdt+ CA1PNs within the pyramidal layer to Figure 4, as is done in most of the other figures, would also be insightful. The cFos measurements in Figure 7 appear to be similarly underpowered.

We are grateful for this valuable input. We added the corresponding analysis and graphs as supplementary material to Figures 4 and 7.

6. While the description of the E12.5 CA1PN pioneer population is intriguing, it is unclear whether this relatively sparse population not following the same trends as between E14.5 and E16.5 neurons convincingly disproves the importance of birth order in predicting CA1PN diversity. For many traits, E12.5 and E16.5 neurons show considerable similarity, suggesting that they may just be early and late emerging populations of deep CA1PNs.

We are a little confused with this last point and apologize if we have not been clear enough in the previous manuscript. Indeed, we agree with the reviewers statement that “For many traits, E12.5 and E16.5 neurons show considerable similarity”, however they are certainly not just “early and late emerging populations of deep CA1PNs” because E16.5CA1PNs are not deep but superficial CA1PNs. In other words, these two populations with different birthdates display many similar intrinsic morpho-physiological features but they are located in different parts of the CA1 layer (and also differ in their cfos labelling). We hope we understood this reviewer’s point correctly.

Reviewer #3 (Recommendations for the authors):This is an interesting study by Cavalieri et al., examining the birth date-specific electrophysiological features, wiring and function of subpopulations of CA1 pyramidal cells of the rodent hippocampus. It presents a set of stimulating facts about neurons fate mapped at E12.5, E14.5 and E16.5 that can have important implications on the role of development on adult function of circuits. I have a number of suggestions that I believe would strengthen the findings of the study that the authors may want to consider.1) The fate-mapping strategy that has been successfully employed by the researchers in this and previous studies allows them to mark cells that are becoming post-mitotic at the three developmental time points that they have selected. Nevertheless because of the limitations of the method, only a handful of cells born at a given time point are labeled and this is especially true for the E12.5 fate-mapped neurons. This precludes the collection of extensive datasets and the opportunity to make robust statements on the stratification of CA1 pyramidal cells based on birth date versus location. One way this limitation could be overcome for two of the main and most novel findings the researchers have obtained, namely the tracing and behavioral/c-fos data, is by utilizing BrdU labeling. The researchers could perform BrdU labeling at the three different embryonic time points, followed by the tracing and c-fos/behavioral experiments to show that the more extensive data they will get matches the ones obtained with Ngn2CreER.

We thank the reviewer for his/her constructive comments. This is a very good suggestion, however, we do not expect the BrdU labeling to reach such a higher yield than the Ngn2CreER experiments. This is particularly true for the early time points (E11-12) and was previously reported (see for example Caviness et al., 1965). In addition, we would need to use transgenic mice in which GABAergic neurons can be separated from PNs, given that E12 is close to their peak of neurogenesis and that they will be labelled with BrdU as well. If absolutely needed, we can repeat the Caviness experiments on GAD67-Cre mice crossed with Ai14, but this would require at least 6 additional months and should not change our conclusions.

2) The representative examples of the recording sEPSC shown in figure 2 are very noisy. It would therefore be of added value if the authors showed examples of which of the peaks are detected as events by the criteria and pipeline they are using.In addition it would be good if they also showed representative examples of the sIPSCs in the figure and reported how they detected peaks for those too.

We apologize for the bad quality of our illustrations and have chosen different representative examples and illustrated sIPSCs as well following the reviewer’s advice.

3) On the bottom of page 12 the authors make the following statement: "Interestingly, none of the intrinsic electrophysiological properties measured here correlated with the location in the stratum pyramidale, suggesting that the embryonic birth date is a major determinant of the observed cell heterogeneity, which cannot solely be explained by the radial gradient".The reviewer finds this to be a strong statement based on the data the authors show. There are two points from the E16.5 group that are closer to the SO and they have overlapping values with the E12.5 group.

We have carefully re-examined the data mentioned by the reviewer and found that the values for the two E16CA1PNs located deeper (close to the so) are perfectly in line with the distribution of values obtained in the rest of the E16 cells, which are not different from the ones for E12.5 neurons. In other words, the fact that the membrane properties of “ectopic” (i.e. E16CA1PNs closer to the SO) are maintained despite their soma location, further supporting our claim that embryonic origin is a stronger determinant than location.

In general, since the authors show that birth date correlates well with positioning, making a claim that the position does not matter, but it is the birth date that does, reads somewhat contradictory.Along the same lines and as it regards figure 2C the authors claim that: in contrast, the frequency of s-EPSCs (Figure 2C) showed a linear correlation with the cell body location (r = 0.49, p: 0.002). This was also reflected in the E/I frequency ratio (r=0.43, P: 0.0047), indicating a higher excitatory synaptic drive in deep than in superficial neurons, regardless of their birthdate (Figure 2E).This statement would suggest that it is the position that determines the features of the inputs and not the birth date.

We apologize for the lack of clarity. Our dataset indeed indicates that birthdate matters more than location for intrinsic properties (membrane properties, calbindin expression, dendritic morphology, etc) but that the opposite is true for sEPSCs or evoked IPSCs. However, we do not show that birth date correlates with positioning as E12 CA1PNs evenly distribute along the radial axis, and since there are some ectopic cells (cf. previous point), that despite their location, display intrinsic properties reflecting their age.

In general, it would be informative, since this one of the main points of the study, if the authors showed which point reported in the graphs corresponds to which cell as it regards its position in CA1. Adding more cells from the tails of the spatial distribution of the cohorts would strengthen the claim that the important factors here is birth date and not position.4) In terms of the data presented in figure 3. The manuscript would be further improved if the authors looked at the integration capacities of the different 3 cohorts of fate-mapped cells using current clamp and SC stimulation. Also, since it is shown that there is a difference in the perisomatic PV-positive inhibitory synapses the E14.5 ones receive, why not perform stimulation of the pyramidal cells layer to record evoked IPSCs coming from these synapses instead of the SC?

We agree with the reviewer that the analysis of the integration properties of the 3 cohorts in response to SC stimulation in current-clamp would be an interesting addition (among quite a few others). In fact, we have started a collaboration with Dr. J Makara’s group to address a similar (but slightly different) question. So far, Dr. Makara’s lab has obtained results describing the dendritic excitability (not the integration of inputs) of the E12 and E16 groups in the dorsal CA1, but they are missing the E14 CA1PNs. If the reviewers and editor find that this is a necessary addition to the paper, at least 6 additional months would be required (for experiments and analysis). They would obviously co-author the paper.

5) In many of the figures (including figure 3) the all the individual data points are presented, but with no mention on the actual number of points and that of biological replicates. It would be good for the reader to have these numbers indicated in the panels of the figure or in the figure legend, since it is important for the statistics applied.

We apologize for this lack of clarity, the number of points and replicates is now mentioned in the text, before the description of each set of results. Supplementary material was added to Figures 4 and 7, in order to better clarify the sample size used in these experiments.

6) Along the same lines, in table 2 the authors report the number of mice used for each of the behavioral experiments with a number of conditions only having "2" animals. I think the results would be more robust if the authors increased the number for each of the conditions to at least "3".

We agree with the reviewer that having more mice is always better, however, for these cfos experiments we have been able to use a total of 23 fate-mapped animals, which, when taking into account the difficulty to have them, is quite a significant number. Adding more mice would require at least 4 months of experiments.

7) Regarding the data presented in figure 5 D1 and D2 on the sag at -200pA. The data here is clear, but it would also be informative to report whether the sag is different if the different groups are hyperpolarized at the same potential rather than using the same magnitude current pulse. It is expected that cells that have a higher input resistance (E14.5) would show a higher hyperpolarization upon the same current pulse. I would suggest that the authors inject enough current to hyperpolarize the cells for each group to the same value and assess if the sag is indeed different between the groups.

We are grateful for the suggestion. However, we respectfully disagree with the reviewer. It is not uncommon to calculate the sag from a current step at a fixed value. In our case, we followed the same protocol as the one used in the Spruston lab (see Graves et al., 2012).

8) In figure 7B2 one would potentially expect that there would be a significant increase in the labeling of cells in NOV over FAM environment, since it has been previously shown that the amount of activation of cells in a more familiar environment decreases compared to a novel one. This would also correlate well with the decreased exploratory behavior of the mice in this experiment. The authors should make a comment on this point and on how their c-fos expression may correlate with prior spiking activity and over which time scales.

We thank you for the input, we adapted the Discussion section accordingly. In addition, we would like to point out that although there is a correlation between the decrease in exploratory behavior (running) and the decrease in cFos expression as an environment becomes more familiar, this relationship is not causal. cFos expression does not merely reflect activity of place cells but rather memory related processes associated with changes in neuronal plasticity (Fleischmann, 2003). In other words, it is difficult to provide a straightforward interpretation relating neuronal spiking directly to c-fos expression.

9) Since one of the most interesting findings of the study is the differential proportion of output nuclei shown for some of the cohorts, it is important that the specificity of the injection sites is presented in the most convincing manner. In this respect the panel in Supplementary figure 1 C targeting the lateral septum is not as clear, since there seem to be many fibers labeled, which also makes it hard to spot where the injection site is. Perhaps adding the respective digital atlas next to each image and a close up insert of the injection site would be helpful for the readers to navigate the position of the injections.

We thank the reviewer for an excellent suggestion. The specificity of the injection sites as well as their position relative to the atlas is now provided.